# The Application of Ethnomedicine in Modulating Megakaryocyte Differentiation and Platelet Counts

**DOI:** 10.3390/ijms24043168

**Published:** 2023-02-05

**Authors:** Fei Yang, Jia Lai, Junzhu Deng, Jun Du, Xi Du, Xiaoqin Zhang, Yiwei Wang, Qianqian Huang, Qian Xu, Gang Yang, Yanjun Zhang, Xilan Zhou, Xiao Zhang, Yuan Yuan, Chunxiang Zhang, Jianming Wu

**Affiliations:** 1School of Pharmacy, Southwest Medical University, Luzhou 646000, China; 2School of Basic Medical Sciences, Southwest Medical University, Luzhou 646000, China; 3School of Pharmacy, Chengdu University of Traditional Chinese Medicine, Chengdu 611137, China; 4Education Ministry Key Laboratory of Medical Electrophysiology, Southwest Medical University, Luzhou 646000, China; 5Luzhou Key Laboratory of Activity Screening and Druggability Evaluation for Chinese Materia Medica, Southwest Medical University, Luzhou 646000, China

**Keywords:** botany, blood system, research progress, megakaryocyte, blood platelet

## Abstract

Megakaryocytes (MKs), a kind of functional hematopoietic stem cell, form platelets to maintain platelet balance through cell differentiation and maturation. In recent years, the incidence of blood diseases such as thrombocytopenia has increased, but these diseases cannot be fundamentally solved. The platelets produced by MKs can treat thrombocytopenia-associated diseases in the body, and myeloid differentiation induced by MKs has the potential to improve myelosuppression and erythroleukemia. Currently, ethnomedicine is extensively used in the clinical treatment of blood diseases, and the recent literature has reported that many phytomedicines can improve the disease status through MK differentiation. This paper reviewed the effects of botanical drugs on megakaryocytic differentiation covering the period 1994–2022, and information was obtained from PubMed, Web of Science and Google Scholar. In conclusions, we summarized the role and molecular mechanism of many typical botanical drugs in promoting megakaryocyte differentiation in vivo, providing evidence as much as possible for botanical drugs treating thrombocytopenia and other related diseases in the future.

## 1. Introduction

Platelets are well known for their powerful roles in vascular injury, inflammation and wound healing [1]. However, megakaryocytes, the precursor cells of platelets, have not been well known until recent decades. Megakaryocytes, a type of hematopoietic cell, undergo a series of complex processes to generate platelets, such as growth, development and maturation. At the same time, megakaryocytes possess certain cellular capabilities of immunity and inflammation [2]. In recent years, various experiments have mutually confirmed that megakaryocytes play an important role in the process of self-renewal and differentiation of HSC into multifunctional lineages [3,4,5]. In addition, as a type of intercellular communication cells, megakaryocytes also maintain the function of niches in the bone marrow [6]. Therefore, megakaryocytes play a very important role in the hematopoietic system.

Megakaryocytes release an average of 1 × 10^11^ platelets into the blood each day. Megakaryocyte differentiation and maturation and platelet production are complex processes regulated by strict factors. This process is a hot topic in the field of hematopoiesis. In most clinical cases, thrombocytopenia is defined as a platelet count less than 100 × 10^9^/L, with clinical manifestations of splenomegaly, purpura, skin and mucous membrane hemorrhage, visceral hemorrhage, intracranial hemorrhage and even death [7]. Thrombocytopenia is a secondary complication of many diseases, such as myelodysplastic syndrome [8], leukemia [9], dengue fever [7], and parasitic infections [10]. In addition, myelosuppression caused by chemoradiotherapy also presents with a decreased platelet count [11]. To treat thrombocytopenia, platelet infusion, cytokines and TPO receptor agonists are commonly used in clinic [12]. However, platelet supply is scarce, the cost is high, the storage time of platelet is short, and it is easily contaminated by bacteria. Bloodborne infections may exist in the transfusion process, and homologous immune-related complications may also be caused. Cytokines stimulate platelet production by increasing the number of hematopoietic stem cells and blood cells. However, cytokines have many adverse reactions, including arrhythmia and acute pulmonary edema [13]. Thrombopoietin receptor agonists currently include first-generation recombinant human thrombopoietin (RhTPO) and second-generation agents (TPO-RAs), such as romiplostim and eltrombopag. Although they promote megakaryocyte differentiation and platelet production in vivo and increase the number of peripheral blood platelets, RhTPO may cross-react with endogenous TPO of patients to neutralize autoantibodies, which may result in immunogenic reactions, while romiplostim is the only TPO-RA that binds with endogenous TPO at the same site without resulting in an immunogenic reaction. However, there is a risk of thrombosis and up to 10% discontinue reaction. Both them require regular treatment and frequent monitoring. [14,15,16]. In recent years, ideas of platelet generation in vitro have been proposed, and many laboratories have successfully differentiated megakaryocytes into platelets in vitro. However, the large-scale production of platelets is still a challenging task in vitro [17].

Although the above treatment agents have significant efficacy, they have many adverse reactions and are limited in clinical application. In view of this, it will be a very important research direction to find long-acting, low-toxicity and inexpensive drugs to promote megakaryocyte differentiation and platelet generation. Natural drugs, represented by botanical drugs, have a long application time, stable efficacy, few adverse effects and a relatively low cost, providing a new direction for promoting platelet generation [18]. We used the terms “botanical medicine” and “herbal medicine”, “differentiation of megakaryocytes” and “platelets” through Google Scholar, PubMed and Web of Science, covering the period 1994–2022. Obviously, based on the data in Figure 1, there are a large number of experimental studies showing that a variety of traditional herbal medicine preparations or monomeric components can promote megakaryocyte differentiation in vitro and in vivo. Here, we focus on the role of herbal medicine in promoting megakaryocyte differentiation and molecular mechanisms in vivo. We hope that research on herbal medicines and their active components in megakaryocyte differentiation will provide references for the research and prevention for related diseases in the future. Therefore, in this article, we review the effects of herbal medicine on megakaryocyte differentiation, thrombocytopenia, leukemia, myelosuppression and other diseases.

## 2. The Process of Megakaryocyte Differentiation

Megakaryocyte differentiation is a complex process in which hematopoietic stem cells differentiate into megakaryocytes, undergo terminal differentiation, and finally form proplatelets. In the classical process of hematopoietic stem cell self-renewal and directional differentiation, long-term hematopoietic stem cells differentiate into multiple multifunctional progenitor cells, then multiple multifunctional progenitor cells differentiate into myeloid progenitors and lymphoid progenitors, myeloid progenitors differentiate into megakaryocyte-erythroid progenitors and granulocyte-macrophage progenitors, and eventually differentiate into megakaryocyte progenitors and erythroid precursors [19]. Megakaryocyte progenitors eventually differentiate into megakaryocytes. After terminal differentiation and maturation, megakaryocytes eventually form proplatelets [20].

However, several recent studies have found that in addition to the ability to self-renew, long-term hematopoietic stem cells (LT-HSCs) can differentiate into short-term hematopoietic stem cells (ST-HSCs), which can differentiate into multiple multifunctional progenitors [21]. They are mainly divided into MPP1, MPP2, MPP3 and MPP4 (Figure 2). MPP1 is classified as a short-term HSC. MPP3 is largely granulocyte/macrophage-biased. MPP4 differentiates into common lymphoid progenitor cells (CLPs), which differentiate into T cells and B cells. MPP2 [22] cells are progenitor cells that tend to differentiate into common myeloid progenitors (CMPs). CMPs can differentiate into granulocyte-macrophage progenitors (GMPs) and megakaryocyte-erythroid progenitors (MEPs). MEPs differentiate into erythroid progenitors (EryPs) and megakaryocytic progenitors (MkPs). At the same time, MPP2 cells could skip the process from HSCs to CMPs and MEPs and directly differentiate into megakaryocytes. Another study reported that approximately 60% of hematopoietic stem cells are VWF+ cells in the bone marrow. In the hematopoietic microenvironment, because megakaryocytes generate platelets, they exist in vascular sinuses, and VWF + HSC have the same transcription factors and cytokines. Megakaryocytes are thought to be predisposed to induce VWF + HSC to differentiate into megakaryocytes and generate platelets [3]. Therefore, we hypothesize that VWF+ cells have a specific tendency to differentiate into myeloid lineage cells under certain conditions. These two pathways can improve the efficiency of megakaryocytes differentiation in the hematopoietic lineage and provide an effective method for rapid platelet production in the hematopoietic system. After that, megakaryocytes will undergo multiple mitoses to undergo polyploid terminal differentiation and maturation, in which the largest polyploid nucleus reaches 128 N. Once megakaryocytes mature, they extend proplatelets into the vascular sinuses through the energy provided by the cytoskeleton and then release platelets through shear forces [23].

## 3. Molecular Mechanisms Affecting Megakaryocyte Differentiation

### 3.1. The Role of Signaling Pathways in Megakaryocyte Differentiation

Megakaryocytes are regulated by transcription factors, adhesion factors, cytokines and chemokines in the hematopoietic system. Among these mediators, thrombopoietin (TPO), the ligand for the c-MPL receptor, is the most important for megakaryocyte generation and differentiation (Figure 3A). TPO is considered to be a key regulator of megakaryocyte differentiation and can initiate a variety of signal transduction pathways, including JAK2/STAT3/STAT5, MAPK/ERK and PI3K/AKT [24,25]. ERK1/2 promotes the differentiation of HSCs into megakaryocytes under continuous activation [26]. PI3K/AKT has been implicated in the late differentiation of megakaryocytes [27]. As a downstream signaling pathway of JAK2, STAT3/STAT5 is also involved in megakaryocyte maturation and platelet generation. Some researchers knocked out the c-MPL receptor in mice and found that the mice still had a small number of platelets left in their blood that kept them alive. However, there is no evidence that TPO is involved in the late differentiation of megakaryocytes and platelet formation.

Therefore, some scholars have made a reasonable guess that the differentiation and maturation of megakaryocytes are regulated by a variety of other factors apart from TPO. Interleukins, such as IL-6, IL-11 and IL-1α, can stimulate megakaryocyte differentiation and promote platelet formation. Vascular endothelial growth factor (VEGF) is secreted by various hematopoietic and nonhematopoietic cells and has three homologous receptor tyrosine kinases (VEGFR1/VEGFR2/VEGFR3) that play roles in different stages of megakaryocyte differentiation [28]. Insulin growth factor (IGF-1) can promote CD34+ cells to differentiate into megakaryocytes and promote platelet formation [29]. Tyrosyl-tRNA synthetase (YRS^ACT^) was shown to induce the differentiation of hematopoietic stem cells in TPO-deficient signaling [30]. Chemokine ligand 5 (CCL5, RANTES) has been shown to bind to CCR5, a receptor of megakaryocytes, to promote megakaryocyte differentiation and proplatelet generation [31]. In addition, human growth hormone (hGH) may exert a complementary and synergistic effect with c-MPL ligands on thrombopoiesis [32]. The discovery of these factors has gradually completed the process of megakaryocyte differentiation in the hematopoietic system, which provides the possibility for the drug treatment of megakaryocyte-related diseases.

### 3.2. The Role of Transcription Factors in Megakaryocyte Differentiation

Megakaryocyte differentiation depends largely on the expression of transcription factors in the nucleus. Various transcription factors play an important role in each stage of HSC differentiation into megakaryocytes (Figure 3B), such as GATA1, FOG1, FLI1, RUNX1, NFE2 and SCL2. GATA1 affects the differentiation of HSCs into megakaryocyte-erythroid progenitors [33]. GATA2 promotes MEP differentiation into megakaryocytes accompanied by erythropoiesis. RUNX1, a master regulator of the hematopoietic system, inhibits the erythroid-specific transcription factor KLF1, thereby changing the ratio of FIL1/KLF1 and promoting megakaryocyte differentiation. RUNX1 may also promote hematopoietic lineage migration to megakaryocytes by downregulating the expression of MYH10, a gene encoding non-muscle myosin heavy chains [4]. Early growth factor (EGR1) can promote MEP differentiation into megakaryocytes [2]. Other transcription factors, such as FOG1, FLI1, ETV6 and GFI1B, can promote megakaryocyte differentiation [34]. VWF+ and NF-E2 also promote the differentiation and maturation of megakaryocytes to generate proplatelets.

Transcription factors play an integral role in megakaryocyte differentiation and platelet release, ensuring a stable number of circulating platelets under stable conditions. However, there are still a large number of transcription factors involved in the complex regulatory process of megakaryocyte differentiation that have not been fully explained.

## 4. Effects of Phytochemicals on Megakaryocyte Differentiation

For a long time, as a natural treasure house of nature, botanical medicine has played an important role in the development of traditional medicine [35,36]. Many botanical drugs are considered to have rich pharmacological effects. Phytomedicine has become one of the most important sources of modern drug discovery because of its low side effects and low cost. This section describes the pharmacological activity and possible molecular mechanisms of botanical drugs in the treatment of thrombocytopenia, leukemia, and myelosuppression by promoting megakaryocyte differentiation (Figure 4). In addition, we summarized some botanical drug active components (Table 1) and compound preparation (Table 2) with certain data support. We expect to find some botanical drugs among them and look forward to their clinical application.

### 4.1. Effects of Phytochemicals on the Treatment of Thrombocytopenia by Promoting Megakaryocyte Differentiation

Since there are many kinds of plant drugs that promote megakaryocyte differentiation and treat thrombocytopenia, we will describe them one by one in terms of botanical drugs, plant monomeric components and derivatives of natural products.

#### 4.1.1. Botanical Drugs

##### *Panax ginseng* Active Ingredients and Compound Preparations

As the king of oriental herbal medicine, *Panax ginseng* C. A. Mey. (PG) is widely found in China, Korea and Japan [37]. It has the effects of tonifying vitality, nourishing the blood, stimulating the body, and tonifying the five viscera. Ginsenoside, one of the most important active components of PG, has various neuroprotective, anticancer and antidiabetic effects. Related experiments have shown that ginsenosides can also alleviate myelosuppression by restoring hematopoietic and immune function [38].

Ginsenosides, a kind of triterpenoid saponin, account for approximately 2%–3% of the total PG content. It has been reported that ginsenosides can promote the proliferation and differentiation of CD34+ cells through the synergistic action of hematopoietic growth factors such as TPO, IL-3, granulocyte colony-stimulating factor (G-CSF) and erythropoietin (EPO). According to the main chain structure, ginsenosides can be divided into dammarane type, oleanolic acid type and ocotillol type. Dammarane sapogenins (DS) are de-glycosylated products of ginsenoside saponins. It can increase the number of PLTs, colony forming unit-megakaryocytes (CFU-Meg) colony forming unit-granulocytes, erythrocytes, monocytes, and megakaryocytes (CFU-GEMM) in the peripheral blood of mice with myelosuppression and effectively relieve the symptoms of myelosuppression in irradiated mice.

Through drug analysis and the separation of components, the ginsenoside diol saponin component (PDS-C) was isolated from the total ginsenosides, which contained five monomeric ginsenoside components. PDS-C was made into “Painengda” capsules, and it was approved as a new class of five proprietary Chinese medicines by the China Food and Drug Administration in 2010. It is used to treat pancytopenia caused by various causes. There are experiments showing that after 7 days of continuous treatment with PDS-C, the peripheral blood platelet levels of CTX-induced myelosuppressive mice recovered significantly in a dose- and time-dependent manner. Histopathological sections of mice showed a significant increase in the number of three hematopoietic lineage cells, especially megakaryocytes, in the bone marrow of the model group compared with that of the PDS-C group. The number of megakaryocyte colony factor (CFU-MK) increased significantly in mice treated with PDS-C. The experimental results show that. PDS-C activates the MAPK pathway through up-regulation of p-ERK and p-MEK and up-regulation of transcription factors such as c-kit and GATA1, which can promote megakaryocyte proliferation and differentiation and restore platelet levels [39]. The experiments of PDS-C intervention on CHRF-288 cells and Meg01 cells showed that PDS-C could increase the proportion of CD42b, CD41, TSP and CD36 in megakaryocytes and increase the expression of GATA1 and RUNX1, indicating that PDS-C has the ability to induce megakaryocyte differentiation in vitro [40].

In addition, there is a rare kind of ginsenoside in *Panax ginseng* C. A. Mey., which is the hydrolysate of the original ginsenosides and is rarely detected in nature. Compared with the original ginsenoside, rare ginsenoside has lower molecular weights and is more easily absorbed by the human body, so it has better pharmacological activities [41]. When *Panax ginseng* C. A. Mey. is used in combination with other traditional Chinese medicines, there is a series of complex physical and chemical reactions, resulting in acidic chemical components that reduce the pH value in the decoction, thereby increasing the content of rare ginsenosides [42]. For example, Ginseng-prepared Rehmannia Root (G-PRR) and *Panax ginseng-Ophiopogon Japonicus* (PG-OJ) compound preparations significantly increased the content of rare ginsenoside (Rk3, S-RG3, etc.). It was found that rare ginsenoside can effectively improve the levels of PLTS, CFU-MEG and TPO in mouse bone marrow, suggesting that rare ginsenosides can promote megakaryocyte differentiation and induce platelet production to alleviate bone marrow suppression and promote hematopoietic recovery in mice [43].

*Panax notoginseng* (Burkill) F. H.Chen is one of the herbs in the Panax pseudo-ginseng plant of the *Araliaceae* family. Saponin, as one of the main components of *Panax notoginseng* (Burkill) F. H.Chen, has anti-inflammatory, antioxidant, neuroprotective, antitumor and other pharmacological activities. The main components include panax notoginosides saponins (PNS), ginsenosides and other monomers. In addition to the hematopoietic pharmacological effects of ginsenoside, the expression of MEK, ERK, AKT and PI3K was up-regulated in PNS-treated K562 cells and Meg01 cells. It was also found that the expression of GATA-1 in CHRF-288 cells and Meg01 cells was increased by PNS treatment. Researchers have suggested that PNS may promote megakaryocyte proliferation and differentiation by up-regulating the MAPK signaling pathway and transcription factors of GATA1 [44,45].

##### Total Saponins and Other Effective Monomers from *Sanguisorba officinalis* L.

*S·officinalis* is called Di-Yu in China, Zi-Yu in South Korea and Japan, and Burnet in Western nations. *Sanguisorbae* Radix, the dried root of *Sanguisorba officinalis* L. or *Sanguisorba officinalis* var, is traditionally used to cool blood, clear heat and heal wounds [46]. Studies have shown that in the absence of exogenous IL-3, a concentration of Di-Yu saponins (DYS) higher than 20 mg/L can promote the differentiation and maturation of megakaryocytes, and DYS up-regulates the expression of IL-3R in megakaryocytes [47]. Therefore, the authors hypothesized that DYS has an IL-3-like effect; that is, DYS regulates the proliferation and differentiation of megakaryocytes by coordinating with stem cell factors (SCFs) and its receptor c-kit to activate its downstream signaling pathway.

Since then, the active components of *Sanguisorba officinalis* L. have been found to promote megakaryocyte formation. For example, two components of ellagic acid, 3,3′,4-tri-O-methylellagic acid-4′-O-β-d-xyloside and 3,3′,4-tri-O-methylellagic acid, were found in the ethyl acetate extract of *Sanguisorba* radix. It was proven that the two ellagic acid components increased the expression of the MK-specific marker CD41 in HEL cells, and Giemsa staining proved that the components induced megakaryocyte differentiation, leading to polyploid formation [48]. Another effective component, 3-O-methylellagic acid and ellagic acid (DMAG) and 3,3′,4′-trimethylellagic acid (TMEA), was also extracted with chromatographic separation technology. DMAG increases the expression of CD41 and the maturation marker CD42b in HEL cells and Dami cells and increase the number of polyploids. DMAG can promote the differentiation of megakaryocytes in a myelosuppression mouse model. TMEA increases the expression of surface markers and ROS levels in HEL cells. The researchers predicted that DMAG and TMEA may promote megakaryocyte differentiation, platelet generation and activation by activating the PI3K/AKT signaling pathway. The researchers innovatively combined network pharmacology with traditional Chinese medicine pharmacology to systematically study the relationship between drugs and diseases and verified the accuracy of network pharmacology prediction through experiments. Therefore, a new idea of the relationship between the target and mechanism of botanical drugs in the treatment of diseases has been developed [49,50,51,52].

##### *Angelica sinensis* and Its Compound Preparations

The dried root of *Angelica sinensis* (Oliv.) Diels are widely used in activating blood circulation, stimulating meridian and treating blood deficiency chlorosis. As an effective component of *Angelica sinensis*, polysaccharide of *Angelica sinensis* (APS) was found to increase the number of megakaryocytes in irradiated mice, which made it close to that of the control group. Some researchers have predicted that APS may promote hematopoiesis and platelet formation by up-regulating the PI3K/AKT pathway. It is worth mentioning that many compound preparations of *Angelica sinensis* can also promote hematopoiesis and platelet production. For example, Danggui Buxue Tang (DBT) is a classical traditional Chinese medicine prescription composed of *Astragalus* and *Angelica* at a ratio of 5:1, which is used to treat blood and vital energy, and it has been found that this prescription may enhance the formation of CFU-MK through TPO-independent pathways. Danggui Sini Decoction (DSD) is a classic recipe for relieving pain, blood deficiency, cold coagulation and other symptoms. DSD consists of seven kinds of Chinese medicine, such as *Angelica sinensis* and *Paeonia lactiflora*. DSD significantly increases the proportion of CD34+ cells in mice with bone marrow suppression and significantly increases the mRNA expression level of TPO in the spleen of mice. Unfortunately, until now, no researchers have further investigated the effects of *Angelica sinensis* and its compound preparations in the stages of megakaryocyte differentiation, nor have they explored the signaling pathways of activating platelet activation [53,54,55,56].

##### *Herba Epimedii* and Its Main Extract Icariin

*Herba Epimedii* is the dried leaf of *Epimedium brevicornu* Maxim., which is used as a great tonic. Icariin, an active component of *Herba Epimedii*, can effectively alleviate myelosuppression, increase the number of MEPs in model mice, and increase the release of hematopoiesis-related factors, such as G-CSF and TPO. We could reasonably speculate the possibility of icariin inducing the differentiation of megakaryocytic progenitors by stimulating the release of TPO, thereby promoting platelet production [57].

##### Red Yeast Rice, Date Palm and Guava Leaf

It is worth mentioning that some Indonesian scholars studied effective drugs promoting megakaryocyte differentiation and platelet formation, discovering that red yeast rice derived from fermented rice by *Monascus purpureus* mold, data palm (*Phoenix dactylifera* L) and guava (*Psidium guajava folia*) can increase platelet levels. They contain flavonoids and isoflavones, which can inhibit hyaluronidase degradation of hyaluronidase to increase the level of IL-6 in the bone marrow extracellular matrix (ECM). IL-6 can stimulate the generation and differentiation of megakaryocytes and can promote the level of platelets in the blood. In addition, quercetin, a flavonoid extracted from guava leaf, could promote the differentiation of stem cells into megakaryocytes by increasing the levels of CM-CSF and IL-3 cytokines in the bone marrow [58].

##### *Justicia adhatoda* L. (Vasika)

*Justicia adhatado* L., also locally known as Vasika and Baikar, has long been used in the Indian Ayurvedic system of medicine. It is widely found in Pakistan, India and Panama. Its leaves are used to treat rheumatism, pneumonia, cough and other related diseases. Dami cells treated with *Justicia adhatoda* L. had typical lobulated nuclei, granular cytoplasm and an enlargement of cells and nuclei. In addition, the expression of CD41 and CD42b was significantly increased. The expression of RUNX1, c-MPL and SOCS in Dami cells treated with *Justicia adhatoda* L. was increased, and megakaryocyte differentiation was induced. Researchers have also found that *Justicia adhatoda* L. induces megakaryocyte maturation and differentiation by enhancing the production of mitochondrial reactive oxygen species and enhancing mitochondrial membrane permeability [59].

##### *Carica* *papaya*

*Carica papaya*, commonly known as papaya, is a tropical tree belonging to the Caricaceae family. Numerous experiments have shown that papaya leaves [60] and seeds [61] can improve thrombocytopenia induced by various conditions, such as Gordon’s fever and chemotherapy [62]. Papaya leaf juice has been confirmed to promote megakaryocyte maturation and differentiation and release platelets by increasing the expression of CD110, a marker on the megakaryocyte surface [63]. We look forward to further research on the molecular mechanism of *Carica papaya* in thrombocytopenia.

#### 4.1.2. Plant Monomeric Components

##### Orientin, a Flavonoid Glycoside from Tulsi

Orientin, a glycosidic flavonoid or dietary administration of Orientin containing Tulsi (Holy Basil) leaves. It is widely used in some traditional Indian medical systems, such as Ayurveda, Unanr and Sida. The researchers treated mice with the powder of its dry leaves and found a significant increase in peripheral blood platelets. Orientin was found to have almost twice as many megakaryocytes as the dry leaves group. Interestingly, the expression of CD41 and CD62P (activated platelet marker) in megakaryocytes was detected to increase by flow cytometry. It was proven that orientin triggered the formation, maturation and differentiation of megakaryocytes and the generation of platelets. Subsequently, the research group also found that orientin could significantly increase megakaryocyte-specific cytokines, including Egr1, CXCL12, IL-6, Fn1, CCL5 and CSF2. However, these transcription factors are related to the development, differentiation and expansion of megakaryocytes [64,65].

##### Hirsutine, a Tetracyclic Heteroyohimbine Alkaloid from Uncaria Rhynchophylla (Miq.) Miq. Ex Havil.

Hirsutine (HS) is a tetracyclic heteroyohimbine alkaloid predominantly found in the *Uncaria* and *Mitragyna* genera. HS enhanced the expression of the specific markers CD41 and CD42b in K562 cells and Meg01 cells. In addition, HS promoted polyploidy in these two cell lines. The researchers subsequently found that the expression of mRNAs, such as GATA1, FOG1, RUNX1 and NFE2, was significantly increased in K562 cells. It was also found that HS promoted megakaryocyte differentiation by activating the MEK/ERK pathway to up-regulate transcription factors such as FOG1 and GATA1 [66].

##### Fucoidan, a Sulfated Polysaccharide from Algae

Fucoidan is a kind of fucose-rich sulphated polysaccharide that has antioxidant and antitumor activities. Most importantly, researchers have found that fucose can stimulate hematopoiesis. For example, a fucoidan polysaccharide (SFF) and its simulated digestion product fucoidan (DSFF) prepared from *Sargassum fusiforme*. can effectively promote platelet recovery in mice with bone marrow suppression induced by cyclophosphamide. DSFF could promote the differentiation of K562 cells and platelet formation by increasing the expression of GATA1 and other transcription factors and alleviate myelosuppression and blood cell abnormalities induced by cyclophosphamide. In addition, sea cucumber fucoidan (*Holothuria Polii*) and brown algae (*Chordaria flagelliformis*) could also promote the differentiation of hematopoietic cells, increase the number of platelets, and restore hematopoietic function in myelosuppressive mice [67,68,69].

##### Abscisic Acid, a Phytohormone from Higher Plants

Abscisic acid (ABA) is a plant hormone that can inhibit plant growth and is known for its ability to induce leaf abscission. It is widely distributed in higher plants. It has been shown to improve conditions of diseases, such as diabetes, atherosclerosis and enteritis. As a megakaryocyte growth-promoting factor that can activate the ERK1/2 signaling pathway through the non-TPO pathway, ABA promotes megakaryocyte differentiation and platelet production and release [70].

##### Proanthocyanidin A1, an Active Compound from Peanut Skin

Peanut skin, the seed coat of *Arachis hypogaea* Linn., is widely used in TCM clinics. It has a hemostatic effect and is commonly used to treat chronic hemorrhagic diseases. It was found that peanut coats and preparations of its related traditional Chinese medicine could increase the number of platelets. Further experiments showed that the peanut coat extract could promote the differentiation and maturation of MKs and finally accelerate the formation of platelets. Proanthocyanidin A1–4 (PS-1–4) in peanut coats were isolated with high-speed countercurrent chromatography (HSCC). Among them, PS-1 had the best effect on megakaryocyte differentiation and thrombocytopoiesis. After PS-1 intervention in Dami cells, the diameter of the cells, the adhesion of the cells, and the number of nuclei increased. After that, the research group screened the targets between PS-1 and thrombocytopenia through network pharmacology and then carried out molecular docking. Finally, they verified that PS-1 promoted the proliferation and differentiation of Dami cells through the JAK2/STAT3 signaling pathway [71].

##### Api7G, an Active Ingredient from Olive Leaves

Olive leaves are produced in Southeast Asia, such as Vietnam and Thailand. In recent years, olive leaves have been used as health nutrition products. It has been found that their effective components can enhance the differentiation of hematopoietic stem cells, among which apigenin 7-glucoside (Api7G) has been found to enhance the expression of CD41 markers in CD34+ cells. Hematopoietic stem cells differentiate into megakaryocytes [72,73].

#### 4.1.3. Derivatives of Natural Products

##### (R)-TEMOSPho, a Derivative of Lysophosphatidyl Choline from the Natural Products Library

2-(Trimethylammonium) ethyl (R)-3-hydroxy-2-stearamidopropyl phosphate (TEHSPho) is a derivative from the natural product library lysophosphatidyl choline. Researchers found that TEHSPho boosted platelet production, but its structure was less stable. The research group has synthesized many TEHSPho derivatives. Among them, (R)-TEMOSPho can induce cell cycle arrest, cell diameter enlargement, cell polyploidy formation, and high expression of the megakaryotic markers CD41 and CD42b in K562 cells and CD34+ cells. The molecular mechanism by which (R)-TEMOSPho induces megakaryocyte differentiation and platelet generation was predicted and verified in experiments. It was found that (R)-TEMOSPho induced megakaryocyte differentiation by up-regulating the MEK/ERK and PI3K/AKT signaling pathways. The researchers then proposed an alternative strategy to combine (R)-TEMOSPho with TPO. They found that (R)-TEMOSPho could not induce the differentiation of HSCs into megakaryocytes in mice, but it could enhance the activity of TPO in megakaryocyte differentiation and thrombogenesis, which means that (R)-TEMOSPho may be able to be used as an effective reagent for thrombogenesis in vitro [74,75].

##### 1-Palmitoyl-2-linoleoyl-3-acetyl-rac-glycerol (3), the Synthetic Product of the Active Ingredient Monoacetyldiglycerides of Deer Antler

In traditional oriental medicine, deer antler has been used as a tonic to treat anemia, anorexia and fatigue for thousands of years. There are a variety of hematopoietic active ingredients extracted from deer antler, such as monoacetylditriglycerides. Researchers have also synthesized four compounds with similar activity to monoacetylditriglycerides to study their ability to stimulate hematopoietic cells in vitro. This compound, 1-palmitoyl-2-linoleoyl-3-acetyl-rac-glycerol, was found to enhance the stimulation of hematopoietic stem cells and megakaryocytes by up-regulating the transcription factor of SDF-1 and promoting its differentiation into megakaryocytes [76,77].

##### Parthenolide, a Plant-Derived Compound from Some Plant

Parthenolide is a plant-derived compound that has powerful anti-inflammatory properties. It was found that parthenolide could enhance platelet production from megakaryoblastic cell lines, Meg01 and Mo7e. It has been shown that increased ROS and decreased NF-κB activity can induce megakaryocyte differentiation after parthenolide treatment and that NF-κB inhibition is almost certainly a leading cause for the thrombopoietic effects of parthenolide observed in megakaryocytes [78].

**Table 1 ijms-24-03168-t001:** Targeting and signaling pathways related to megakaryocyte differentiation by botanical drug active components in vitro and in vivo.

Ingredient	Part Used	Botanical Name	Genus	Research System	Usage and Dosage	Mechanism of Action	Ref.
PDS-C	Root	*Panax ginseng* C. A. Mey.	Araliaceae	Mice CHRF-288 cells and Meg01 cells	20, 40 and 80 mg/kg 10, 20, 50 mg/L	Up-regulating the MEK/ERK signaling pathway and transcription factors of GATA1 and RUNX1	[40,41]
PNS	Root	*Panax notoginseng* (Burkill) F. H. Chen	Araliaceae	K562 cells Meg01 cells	50 mg/L	Up-regulating the MAPK signaling pathway and transcription factors of GATA1	[45,46]
DYS	Root	*Sanguisorba officinalis* L.	Rosaceae	Baf3 cells 32D cells	10 mg/L	By coordinating with stem cell factors (SCFs) and its receptor c-kit to activate its downstream signaling pathway	[47]
DMAG				Mice HEL cells	5 mg/kg 10, 20 and 40 µM	Activating the PI3K/AKT signaling pathway	[50]
TMEA				HEL cells	10 µM and 20 µM	Activating the PI3K/AKT signaling pathway and increasing ROS levels in cells	[52]
APS	Root	*Angelica sinensis* (Oliv.) Diels	Apiaceae	Mice	2.5 mg	Activating the PI3K/AKT signaling pathway	[54]
Icariin	Aboveground part	*Epimedium brevicornu* Maxim.	Berberidaceae	Mice	10 mg/kg	Inducing the expression of hematopoietic cytokines G-SCF and TPO	[57]
Quercetin	Leaf	*Psidium guajava* L.	Myrtaceae	White rats	108 mg/kg	Hyaluronidase activity is inhibited, and hyaluronidase can release IL-6 and promote megakaryocyte differentiation	[58]
Vasika extract	Leaf	*Justicia adhatado* L.	Acanthaceae	Dami cells	10, 20 and 40 µg/mL	Enhances the generation of mitochondrial ROS and enhances the permeability of the mitochondrial membrane	[59]
Papaya extract	Leaf	*Carica papaya*	Caricaceae	Wistar rats	200 mg/kg	The expression of CD110 on megakaryocyte surface is increased	[63]
Orientin	Leaf	*Ocimum sanctum* L.	Lamiaceae	Mice	1 mg/kg	EGR1, CXCL12, IL-6ST, Fn1, CCL5 and other genes were up-regulated	[64,65]
Hirsutine	Whole plant	*Uncaria rhynchophylla* (Miq.) Miq. Ex Havil.	Rubiaceae	K562 cells Meg01 cells	2.5, 5, 10, 15, and 20 µM	Activating the MEK-ERK-FOG1/TAL1 signaling pathway	[66]
Proanthocyanidin A1	Peanut skin	*Arachis hypogaea* Linn.	Fabaceae	Mice Dami cells	25 mg/kg and 50 mg/kg 20 µM	Binding to JAK2 activates the JAK2/STAT3 pathway and induces megakaryocyte differentiation	[71]
Api7G	Leaf	*Canarium* *Album* (Lour.) DC.	Burseraceae	CD34+ cells	5 µM	It can enhance the expression of CD41 marker in CD34+ cells	[72,73]
Diosgenin	Root	*Dioscorea hispida* Dennst	Dioscoreaceae	HEL cells	10 µM	Promotes the overexpression of p21 and cyclin	[79]

**Table 2 ijms-24-03168-t002:** Plant drug species, associated diseases and prescription sources of the compound preparation in vivo.

Compound Preparation	Plant Name	Diseases	Research System	Usage and Dosage	Prescription Source	Ref.
DSD	Seven kinds of Chinese medicine, such as *Angelica sinensis* (Oliv.) Diels, *Cinnamomum cassia* Nees ex Blume, *Paeonia lactiflora* Pall., *Asarum heterotropoieds* F. Schmidt, *Tetrapanax papyrifer* (Hook.) K. Koch, *Zizypgus vulgaris var. intermis* Bunge and *Glycyrrhiza uralensis* Fisch. ex DC.	Myelosuppression due to chemoradiotherapy	Mice	100, 300, 900 mg/kg	Traditional Chinese medicine	[56]
FEJ	*Equus asinus* L., *Panax ginseng* C. A. Mey., *Rehmannia glutinosa* (Gaertn.) DC., *Codonopsis pilosula* (Franch.) Nannf. and *Crataegus pinnatifida* Bunge	Myelosuppression due to chemoradiotherapy	Mice	27.5, 55 and 110 mg/kg	Traditional Chinese medicine	[80,81]
SYKT	*Zingiber officinale* Rosc., *Daemonorops draco* (Willd.) Blume., *Disoscorea opposita* Thunb., *Poria cocos* (Schw.) Wolf, *Amomum villosum* Lour., *Angelica sinensis* (Oliv.) Diels, *Panax noto-ginseng* (Bur-kill) F. H. Chen and *Glycyrrhiza uralensis* Fisch. ex DC.	Myelosuppression due to chemoradiotherapy	Mice	1.2 g/kg	Dai nationality in Southwest China	[82]
Wei Gan Li	*Cervus nippon* Temminck, *Astragalus membranaceus* Moench, *Panax ginseng* C. A. Mey., *Epimedium brevicornu* Maxim., *Curculigo orchioides* Gaertn., *Cnidium monnieri* (Linn.) Cuss., *Paeonia lutea* Delavay ex Franch and *Glycyrrhiza uralensis* Fisch. ex DC.	Myelosuppression due to chemoradiotherapy	Mice	100 mg/mL, 50 mg/mL and 25 mg/mL	Traditional Chinese medicine	[83]
VITA PLAT	*Carica papaya*, *Cissampelos pareira var. hirsute* (Buch. Ex DC) Forman, *Boerhavia diffusa* L., *Tinospora cordifolia* (Willd.) Miers, and *Azadirachta indica* A. Juss.	Thrombocytopenia due to dengue fever	Wistar rats	200 and 400 mg/kg	Traditional Indian medicine	[84]

### 4.2. Monomeric Plants Can Promote the Differentiation of Erythroid Leukemia into Megakaryocytes and Provide a New Idea for the Treatment of Leukemia

Erythroid leukemia is a type of leukemia in which both red and white cells proliferate and develop into acute myelogenous leukemia [85]. It has been found that differentiation of the hematopoietic lineage into megakaryocytes may be a new treatment [86]. Differentiation therapy is proposed as a mild treatment that induces the differentiation of primitive cells into specific cells and inhibits the proliferation of tumor cells [19]. It has been found that some natural drugs can induce the differentiation of K562 cells (erythroleukemia) and HL-60 cells (myeloblastic leukemia) into megakaryocytes and inhibit the malignant proliferation of K562 and HL-60 cells into erythrocytes and granulocytes. For example, a compound from *Dendrostellera lessertii* was found to induce the differentiation of K562 cells into megakaryocytes, thus restraining the malignant proliferation of leukemia cells to fight leukemia [87].

#### 4.2.1. Nobiletin, a Monomeric Component from Citrus Peel

Nobiletin (NOB), a monomeric component of polymethoxyflavone (PMF), is abundant in the peel of citrus fruits and has been reported to have a wide range of biological activities, including anti-inflammatory, antiatherosclerotic and antitumor activities. Researchers found that NOB could promote the differentiation of K562 cells into megakaryocytes. Therefore, it was found that after K562 cells were treated with NOB, the mRNA expression levels of megakaryocyte surface markers (CD41, CD42b and CD61) were significantly increased, and the mRNA expression levels of the red blood cell marker genes glycoprotein A and hemoglobin α were decreased. In addition, NOB was observed to induce the expression of megakaryocyte differentiation markers in HEL cells, a mature human erythroleukemic cell line. Later, to explore the mechanism of megakaryocyte differentiation, researchers used genome-wide microarray analysis and experimental verification to conclude that NOB may promote the differentiation of megakaryocytes by activating the MAPK/ERK pathways to enhance the expression of EGR1. [88].

#### 4.2.2. A1541-A1543, Limonoid Compounds from the Plant Melia Azedarach

The fruit and bark isolated from the plant *Melia azedarach* have recently been used to treat cancer in traditional Chinese medicine. Three compounds with structures related to limonin were isolated from this tree, namely, the novel limonin compounds A1541, A1542 and A1543. The intervention of A1541 and A1543 on HEL cells induced an increase in the expression of megakaryocyte-specific markers CD41 and CD61. These two compounds have been shown to promote the differentiation of megakaryocytes and inhibit leukemia in animal models by inducing the ERK1/2 pathway [89].

#### 4.2.3. KCR, a Natural Small Molecule from Cyclocarya Paliurus

*Cyclocarya paliurus* (C. *paliurus,* Baral.) Iljinskaja., an endemic plant belonging to the *Cyclocarya* genus of the Juglandaceae family, is mainly distributed in Southern China. It has antiobesity, antioxidation, antidiabetes and other pharmacological effects. An active ingredient, kaempferol-3-O-α-L-(4′-E-p-coumaroyl) rhamnoside (KCR), was isolated from *Cyclocarya paliurus*. It was shown that KCR could induce an increase in the expression of the megakaryocyte marker CD41 in K562 cells and HEL cells and increase the mRNA expression of GFI1B, which proved that KCR could induce erythroleukemia cells to differentiate into megakaryocytes. In addition, the expression of GATA1 was significantly increased in K562 cells and HEL cells treated with KCR, and the expression of p-PKC and p-ERK was up-regulated. This proved that KCR can inhibit the proliferation of abnormal cells by inducing megakaryocyte differentiation and cell cycle arrest through the PKCδ and ERK1/2 signaling pathways, which provides a new therapeutic target for the treatment of erythroid leukemia [90].

#### 4.2.4. Securinine, an Alkaloid from the Genera Securinega, Phyllanthus, and Flueggea

Securinine, an alkaloid, has been found in the leaves of the genera *Securinega*, *Phyllathus* and *Flueggea*. Researchers have explored the possibility that securinine derivatives can induce leukemic cell differentiation. SN3-L6 induces HL-60 cells to differentiate into megakaryocytes with increased cell diameter and nuclei and release platelets. This is a new and unreported differentiation pathway, namely, SN3-L6 induces AML cells to transdifferentiate into megakaryocytes. The discovery of this transdifferentiation pathway provides evidence for the origin of HL-60 cells and megakaryocytes, and its potential value should be worth studying. This study provides a new idea for the subsequent treatment of thrombocytopenia and leukemia. In addition, K562 cells also showed obvious morphological changes after SN3-L6 intervention, and it was also shown that the CD41 and CD61 surface markers of K562 cells were increased via flow cytometry detection, but no platelets were found. Megakaryocytes disappeared on the 5th day after SN3-L6 intervention, followed by K562 cell contraction and apoptosis. This suggests that SN3-L6 may be able to effectively inhibit the malignant proliferation of leukemia cells by inducing the differentiation of erythroleukemia cells into megakaryocytes and apoptosis [91].

#### 4.2.5. Diosgenin, a Steroidal Saponin That Can Be Found in Several Plant Species

Diosgenin, found in fenugreek seeds (*Trigonella foenum-graecum*) and wild yam roots (*Dioscorea villosa*), was found to induce megakaryocyte differentiation of HEL and K562 cells by promoting the overexpression of p21 and cyclin [79].

### 4.3. Phytochemicals That Improve Bone Marrow Suppression and Promote Platelet Production

At present, some drugs have been found to improve bone marrow suppression effectively, especially compound preparations, and evidence shows that they may be related to megakaryocyte differentiation, but the mechanism has not been thoroughly studied.

The small molecule fraction (SMF) of *Polygoni multiflori* Radix Praeparata (PMRP) can improve the mice anemia model induced by cyclophosphamide. It can accelerate the recovery of peripheral blood and increase the number of megakaryocytes in the spleen [92]. Radix *Astragali*, an important traditional Chinese medicine, has been used for hundreds of years to promote hematopoiesis. Astragalus polysaccharide (ASPS), the active compound of Astragalus, also plays a key role in hematopoiesis. ASPS effectively promotes the levels of red blood cells, white blood cells and platelets in mice with bone marrow suppression, and compared with the control group, the number of trilineage hematopoietic cells increased significantly in the ASPS group, especially in megakaryocytes. Researchers hypothesized that ASPS may promote platelet levels by mediating TPO-independent pathways [93].

*Fufang E’jiao Jiang* (FEJ) can improve the hematopoietic function of mice with bone marrow suppression induced by radiotherapy and chemotherapy, and the number of white blood cells and platelets is increased, but no further study has been conducted [80,81]. Sanyang Xuedai (SYKT) is an ancient prescription from the Dai ethnic group in Southwest China, which is composed of eight kinds of Chinese herbs, such as sanguis draconis, radix et rhizome notoginseng, etc. SYKT significantly increases the number of PLTs in mice with bone marrow suppression. However, no further study has been conducted on the related mechanism [82]. Weiganli, a traditional Chinese medicine compound, has been found to significantly increase CFU-Meg and platelet counts at low doses, and it can promote the proliferation of bone marrow mesenchymal stem cells. The proliferation of mesenchymal stem cells is conducive to the recovery of the bone marrow hematopoietic microenvironment and then increases the number of megakaryotic progenitors in the bone marrow, promoting megakaryocyte differentiation and platelet production [83]. A polyherbal formulation (PHF) composed of *Carica papaya*, *Cissampelos pareira*, *Boerhavia diffusa*, *Tinospora cordifolia* and *Azadirachta indica* has been found to treat thrombocytopenia caused by chemoradiotherapy or dengue fever, and its long-term use may aid in early recovery from disease with low platelet counts [84].

These drugs are basically verified to promote the differentiation and maturation of megakaryocytes, generate platelets, and alleviate the body’s myelosuppression. It is worth thinking that researchers have not further verified the way through which drugs promote platelet generation in vitro and in vivo. Their complex components may be the reason why researchers have not conducted further studies. However, it must be said that these drugs can effectively improve bone marrow suppression caused by chemoradiotherapy, and the follow-up study and clinical significance of these drugs are still worthy of expectation.

## 5. Conclusions and Prospects

Although megakaryocytes are the least abundant among hematopoietic cells, they release an average of 1 **×** 10^11^ platelets into the blood per day. Therefore, megakaryocytes differentiation and maturation and platelet release are regulated by strict factors, and this process is also a research hotspot in the hematopoietic system. The laboratory [94] definition of 100 **×** 10^9^/L is thrombocytopenia, clinical manifestations of purpura, visceral bleeding and even death. Thrombocytopenia can be caused by leukemia, dengue fever, malaria, metastatic cancer of the bone marrow and myelodysplastic syndrome, and myelosuppression caused by radiotherapy and chemotherapy. Platelet infusion and TPO receptor agonists [95] are considered common treatments for ameliorating thrombocytopenia, but they can cause harmful thrombosis and immune complications, also, they are costly, and have limitations in their use. However, plant medicine has the advantages of short research, development time and low cost. In Southeast Asia and Africa, traditional herbal medicines and compound preparations are widely used in the treatment of myelosuppression, which may restore the bone marrow microenvironment by promoting megakaryocyte differentiation and platelet formation. In addition, some scholars have found that plant drugs can specifically differentiate MEPs into megakaryocytes and inhibit the malignant proliferation of red blood cells, thus treating patients who are intolerant to chemoradiotherapy and bone marrow transplantation.

We reviewed many botanical drugs and discussed their effects on megakaryocyte differentiation through deeper mechanistic studies. For example, PDS-C, NOB, HS, A1541, A1543, ABA and KCR promoted megakaryocyte differentiation by activating the MEK/ERK signaling pathway. DAMG and APS promoted megakaryocyte differentiation by activating the PI3K/AKT signaling pathway. PS-1 promotes the proliferation and differentiation of megakaryocytes through the JAK2/STAT3 signaling pathway. Some monomers can also activate multiple signaling pathways at the same time; for example, (R)-TEMOPho and PNS activate both the MEK/ERK and PI3K/AKT signaling pathways to induce megakaryocyte differentiation. Rare ginsenoside, DSD and Epimedium may promote megakaryocyte differentiation by increasing the level of TPO. DYS promotes megakaryocyte differentiation by activating the c-kit receptor. Orientin promotes the differentiation of megakaryocytes by stimulating non-TPO signaling. DBT promotes hematopoiesis through a nonclassical pathway. We also found that some botanical drugs can promote megakaryocyte differentiation in other ways. For example, vasika and parthenolide enhance mitochondrial permeability by promoting the generation of mitochondrial reactive oxygen species (ROS), thereby inducing the maturation and differentiation of megakaryocytes. In addition to the abovementioned effects of megakaryocyte differentiation on platelet generation and the partial recovery of hematopoietic function, some drugs have been found to effectively promote MEP differentiation into megakaryocytes, and this differentiation bias may be one of the ways to treat erythroleukemia patients in clinical practice. For example, NOB, A1541, A1543, KCR and securinine can effectively promote the differentiation of dual potent progenitors and erythroid progenitors into megakaryocytes, such as HEL cells and K562 cells, and effectively alleviate the abnormal increase in erythroid cells in leukemia patients by inhibiting their cycle arrest. Previous studies have found that MEPs can preferentially differentiate into the next level of single potent progenitors according to the length of the cell cycle. Coincidentally, the cell cycle of progenitors with megakaryocytic differentiation bias is longer than that of progenitors with erythroid differentiation bias [96,97]. This may be one of the bases for the feasibility of MEP differentiation into megakaryocytes in the treatment of erythroleukemia. Of course, we are looking forward to a more detailed study in the field of megakaryocyte differentiation promoted by plant drugs, opening a new frontier in the treatment of leukemia. In addition, we found a novel differentiation way by which securinine promotes megakaryocyte differentiation, which is the process by which HL-60 granular cells differentiate into megakaryocytes and produce platelets. This novel transdifferentiation pathway needs further exploration. We found that botanical drugs promote platelet production and ameliorate thrombocytopenia caused by myelosuppression. Unfortunately, most botanicals are polypill formulations, and the effects, mechanisms and efficacies of these drugs in promoting megakaryocyte differentiation still have many problems to be solved. For example, the active ingredients of compound preparations are difficult to confirm, and the chemical reactions and physical reactions produced by medicine preparations are difficult to master; however, the pharmacological effect of these compounds in the clinical treatment of myelosuppression should not be ignored. Alternatively, we can try network pharmacology, biology information engineering, and the combination of traditional pharmacology using computers instead of experiments to understand the effects of these drugs on megakaryocyte differentiation through cumbersome big data analysis to make these plants more persuasive for application in clinical medicine.

Natural medicines are plant medicines that have been modified and successfully used to treat a variety of diseases. According to the WHO, herbal medicines are widely used in many countries for healthcare, disease prevention and treatment [98,99]. However, the pharmacodynamics of most botanical drugs promoting megakaryocyte differentiation have not been studied in human patients, so there is a lack of certain clinical research evidence, which leads to a low clinical conversion rate of botanical drugs. Although some drugs, especially compound preparations, have been used in clinical studies, the specific mechanisms by which these herbal prescriptions promote megakaryocyte differentiation and induce bone marrow hematopoietic recovery remain unclear. Therefore, we should further evaluate whether botanical drugs could promote megakaryocyte differentiation in the treatment of related blood diseases to apply them in a more reasonable way through real preclinical studies. Therefore, it is very important to find more potential therapeutic agents to promote megakaryocyte differentiation, improve thrombocytopenia, and enrich the bone marrow environment [100].

Overall, megakaryocytes are special large cells in platelet production; at the same time, megakaryocytes can also affect biased differentiation of HSCs in the hematopoietic system, and the process of megakaryocyte differentiation is cumbersome and complex, in which multiple signaling pathways and a variety of cytokines are involved in the bone marrow microenvironment [101]. Traditional plant ingredients and compound preparations can promote megakaryocyte differentiation and induce its role in the hematopoietic system, which is conducive to the clinical treatment of thrombopenia and myelosuppression. At the same time, we are also looking forward to its therapeutic effect in erythroleukemia.

## Figures and Tables

**Figure 1 ijms-24-03168-f001:**
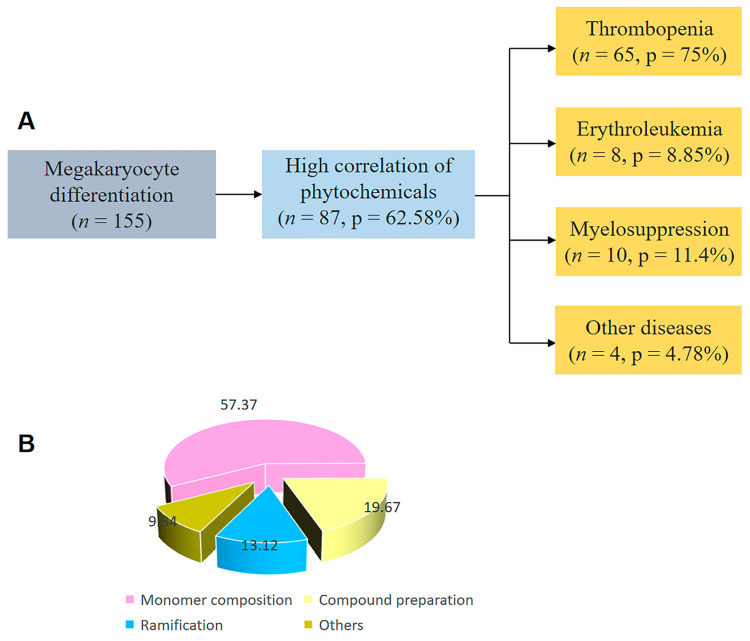
Articles related to megakaryocyte differentiation from the PubMed database. (**A**) The literature on some diseases of megakaryocyte differentiation searched in PubMed. (**B**) The search results were systematically classified into several specific classifications of botanical drugs.

**Figure 2 ijms-24-03168-f002:**
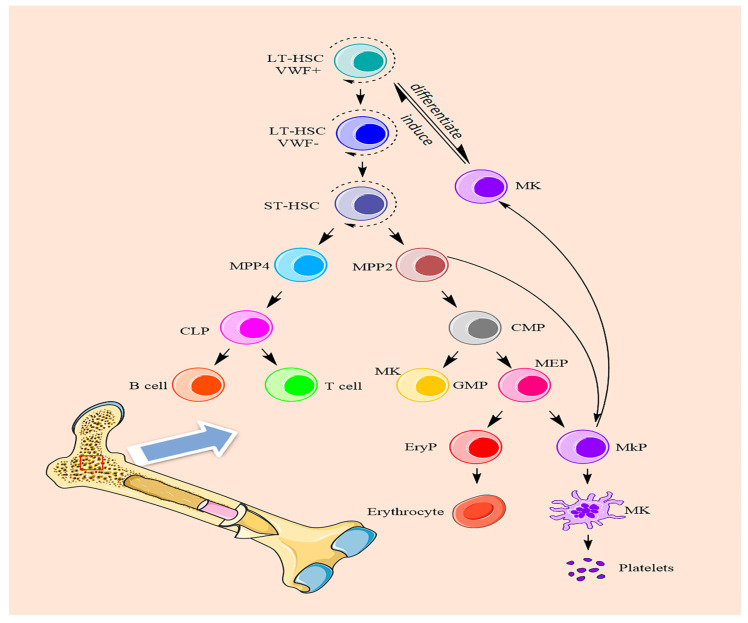
The differentiation processes of megakaryocytes.

**Figure 3 ijms-24-03168-f003:**
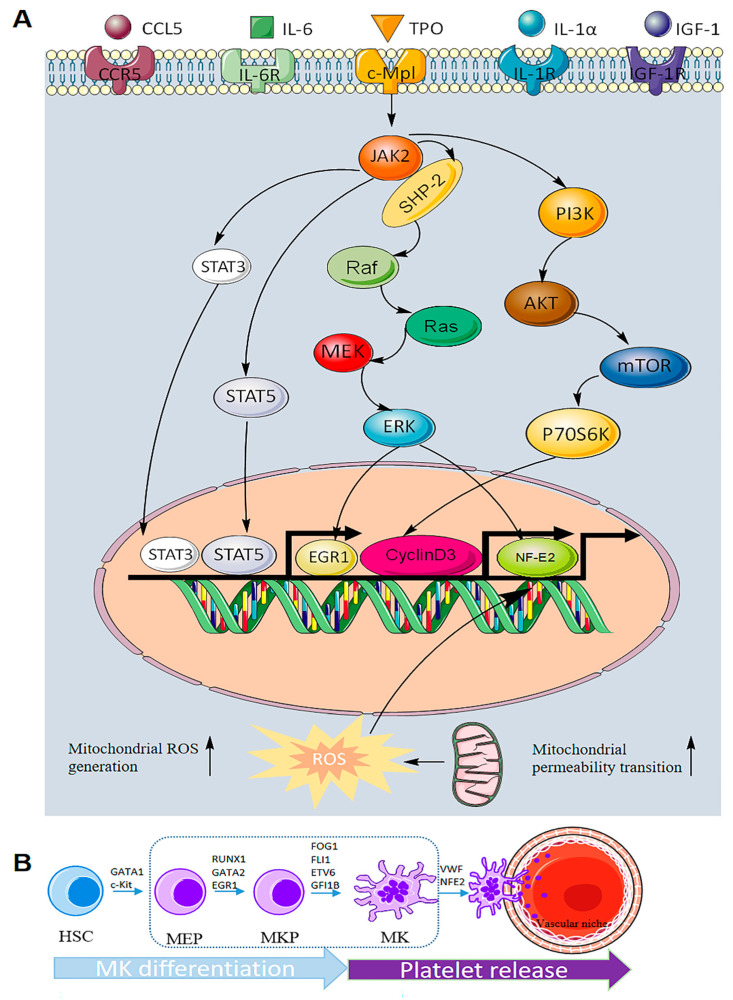
Molecular mechanism of megakaryocyte differentiation. (**A**) The role of signaling pathways in megakaryocyte differentiation. There are at least five regulatory factors in megakaryocyte differentiation, the most important of which is TPO, which mainly initiates three signaling pathways, namely, JAK2/STAT3/STAT5, MAPK/ERK and PI3K/AKT. These intracellular factors related to pathway regulation can promote megakaryocyte differentiation and the generation of platelets. In addition, increased mitochondrial permeability and ROS can also promote megakaryocyte differentiation. (**B**) Transcription factors that play a role in each stage of MK differentiation and platelet release.

**Figure 4 ijms-24-03168-f004:**
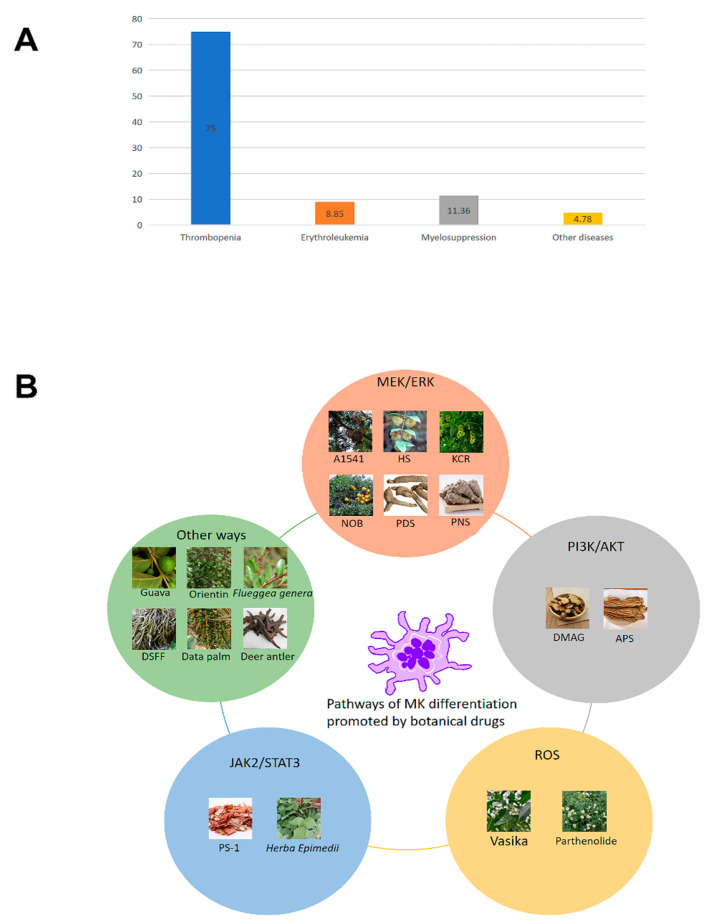
Diseases and molecular mechanisms of megakaryocyte differentiation. (**A**) The literature on some diseases of megakaryocyte differentiation searched in PubMed. (**B**) PDS, NOB, HS, A1541, PNS and KCR promote megakaryocyte differentiation by activating the MEK/ERK signaling pathway. DMAG and APS promote megakaryocyte differentiation by activating the PI3K/AKT signaling pathway. PS-1 and *Herba Epimedii* promote the proliferation and differentiation of megakaryocytes through the JAK2/STAT3 signaling pathway. Vasika and Parthenolide enhance mitochondrial permeability by promoting the generation of ROS. Some botanical drugs promote megakaryocyte differentiation in other ways. DSFF may promote megakaryocyte differentiation by increasing the level of TPO, and orientin promotes the differentiation of megakaryocytes by stimulating non-TPO signaling.

## Data Availability

Not applicable.

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
