# Peer review of "The Application of Ethnomedicine in Modulating Megakaryocyte Differentiation and Platelet Counts"

_ijms, 2023, doi:10.3390/ijms24043168_

Round 1

Reviewer 1 Report

The manuscript of Fei Yang et al. is a review dealing with what is know about compoments from plant origin and megacaryopoiesis as a sort of state of art .The subject is of a great interest and has many implication for future drug developpements and target therapies. However , the paper suffers from many weakness that need complete reformulation

introduction in too long, focused only on importance of targeting platelets production and differenciation  at the difference of  other myeloid lineage . Explain also what is known about megacaryocytopoiseis in myelodysplastic syndrome : that is a unsloved clinical problem. In this pathologies ( low grade myelodysplastic syndrom) when thrombocytopenia is present TPO agonist were tried but with mitiged success and may enforce the importance of the review. in the  introduction  the main question rises only in page 8 of the pdf Since the authors wrote sections of introduction and MK diffrenciation  the true introduction end page 8.  . if the authors want to explain megacaryotyte lineages from stem cells models to progenitors they have to make in a large  separate section ,  focused on mains points and be more precise and in the description of platelet release in vessels ( anatomy not only in the bone marrow) The figure 2: the arrow must be changed since the hematopoiesis is not in the medulla of the bone but in epiphyse  in adult .  a vertebra is a better  bone as an example

after introduction and megacaryopoiese sections, come the first section of plants of interest in megacaryoiesis : those who the mechanisms of action  is supported by animal studies / molecular studies. This section ( 4) is quite clear in term of organization  but may be illstrated with two separate tables : one for the plant drugs effect is demonstrated and active components identified and one for plant drug effect is supported by experiments  but active components are putatitve or in combination 

the study focused mainly on chinese plants (plants that could also be found  either in the world as sanguisorba sp.) is there non chinese species  that could be of interest?

Other major point :

Botany review work  : Taxonomy is important :  the plants names have to be written according to the APG IV classification (2016)  and add the family name to the species name (s) ,  both in the table and in the text; current names only in barkets since they are numerous.  the part of the plant ( root , leave bark....) where the active component is extract must be  shown , it is a part of the review  work.

The section of putative drugs my have to be strongly reduced and the conclusion section also

 the main key messages  will be in the 2 tables with the correct references

there are many English spelling mistakes and unusual style for a science article  as "searchers founds " or the authors says .. instead  of the study by "xxx et al. " or the team of xxxx: more personnalisation is needed and more scientific style is needed . 

Author Response

Jan. 23, 2023

Dear Expert Reviewer,

Thank you very much for the prompt review process and excellent comments. We greatly appreciate the time and efforts which you have spent on it. We are submitting the revised manuscript entitled “Recent progress in the influence of ethnomedicine on megakaryocyte differentiation” (ID: ijms-2097803) to International Journal of Molecular Sciences.

We have carefully considered your comments and suggestions, and addressed each of the concerns in response to the comments (see point by point response). We have revised the manuscripts based on your comments and carefully checked throughout the manuscript and corrected the language errors. Our point-by-point responses to the comments (in blue) are shown below (in red).

  1. Introduction in too long, focused only on importance of targeting platelets production and differentiation at the difference of other myeloid lineage. Explain also what is known about megacaryocytopoiseis in myelodysplastic syndrome and may enforce the importance of the review.

Response: Thanks for your scientific comment, which is very helpful to me. In order to get to the topic of review as soon as possible, namely the application of botanical drugs to megakaryocyte differentiation and platelet generation, I have refined the content of introduction. The process of megakaryocyte differentiation is abbreviated in introduction and described in detail in the section (2). The main clinical accepted treatment of thrombocytopenia and its advantages and disadvantages were briefly described. The advantages of increased use of botanical drugs compared to the above treatment modalities.

  1. Explain what is known about megacaryocytopoiseis in myelodysplastic syndrome : that is a unsloved clinical problem,and may enforce the importance of the review.

Response: Thanks for your scientific comment. Myelodysplastic syndrome (MDS) is a group of hematopoietic stem cell diseases. Hematocytopenia and abnormal development of bone marrow cells are the criteria for confirmation and diagnosis of MDS [1]. Thrombocytopenia is one of the most common causes of death in MDS patients, and thrombocytopenia is more common in low-risk myelodysplastic syndrome (MDS) [2]. The pathogenesis is related to impaired differentiation of megakaryocytes, hyperapoptosis and increased platelet destruction. Currently, the main therapeutic methods include platelet infusion and novel thrombopoietin (TPO) receptor agonists [3]. Platelet infusion has risks such as fever, allergy, infusion tolerance and infusion dependence. TPO receptor agonists can not only stimulate the generation of megakariocytes, but also promote the generation of other hematopoietic cells, leading to the increase of patients' original cells and other adverse reactions. MDS patients with thrombocytopenia have a low survival rate. As a class of drugs that can promote megakaryocyte differentiation, increase thrombocytopenia, improve thrombocytopenia and have low toxicity, plant drugs can be applied in the treatment of MDS, which will be a new research hotspot.

Literature review found that in addition to self-induced thrombocytopenia, thrombocytopenia is also a secondary complication of many diseases, such as myelodysplastic syndrome [1], leukemia [4], dengue fever [5], parasitic infections [6], etc. In addition, myelosuppression caused by chemoradiotherapy also presents with decreased platelet count [7]. We add the above sentence to the introduction, defines thrombocytopenia with platelets less than 100×109/L and describes the specific clinical manifestations of thrombocytopenia.

References:

  1. Garcia-Manero, G.; Chien, K.S.; Montalban-Bravo, G. Myelodysplastic syndromes: 2021 update on diagnosis, risk stratification and management. Am J Hematol 2020, 95, 1399-1420, doi:10.1002/ajh.25950.
  2. Mirnal, M.P.; Terra, L. Myelodysplastic syndrome/myeloprolifertive neoplasm overlap syndromes: a focused review. The American Society of Hematology 2020, 461-464, doi:10.1182/hematology.2020000163.
  3. Hasserjian, R.P. Myelodysplastic Syndrome Updated. Pathobiology 2019, 86, 7-13, doi:10.1159/000489702.
  4. Repsold, L.; Pool, R.; Karodia, M.; Tintinger, G.; Joubert, A.M. An overview of the role of platelets in angiogenesis, apoptosis and autophagy in chronic myeloid leukaemia. Cancer Cell Int 2017, 17, 89, doi:10.1186/s12935-017-0460-4.
  5. Lien, T.S.; Chan, H.; Sun, D.S.; Wu, J.C.; Lin, Y.Y.; Lin, G.L.; Chang, H.H. Exposure of Platelets to Dengue Virus and Envelope Protein Domain III Induces Nlrp3 Inflammasome-Dependent Platelet Cell Death and Thrombocytopenia in Mice. Front Immunol 2021, 12, 616394, doi:10.3389/fimmu.2021.616394.
  6. Zhu, M.; Rong, X.; Li, M.; Wang, S. IL-18 and IL-35 in the serum of patients with sepsis thrombocytopenia and the clinical significance. Exp Ther Med 2020, 19, 1251-1258, doi:10.3892/etm.2019.8347.
  7. Yang, S.; Che, H.; Xiao, L.; Zhao, B.; Liu, S. Traditional Chinese medicine on treating myelosuppression after chemotherapy: A protocol for systematic review and meta-analysis. Medicine (Baltimore) 2021, 100, e24307, doi:10.1097/MD.0000000000024307.
  8. Explain megacaryotyte lineages from stem cells models to progenitors in a large separate section, focused on mains points and be more precise and in the description of platelet release in vessels. Figure 2: the arrow must be changed since the hematopoiesis is not in the medulla of the bone but in epiphyse in adult.

Response: Thanks for your careful review. In the section (2) of this review, the differentiation of stem cells into megakaryocytes. We increased the process of megakaryocyte differentiation and maturation to release platelets in blood vessels, and the diagram of platelet release into blood vessels is added in Figure 3B. According to the literature [1]. bone marrow, as the main hematopoietic organ of the human body, has strong hematopoietic ability mainly in the end of long bone, flat bone and irregular bone, namely the spongy mesh. After understanding the distribution of bone marrow in the body and considering the regular shape of the long bone, namely the femur, it was finally decided to use the femur with obvious epiphyseal structure as the bone model, and the location of the bone marrow with hematopoietic capacity was indicated in Figure 2 (please see in the attachment WORD file).

References:

  1. Morrison, S.J.; Scadden, D.T. The bone marrow niche for haematopoietic stem cells. Nature 2014, 505, 327-334, doi:10.1038/nature12984.
  2. You may be illstrated with two separate tables: one for the plant drugs effect is demonstrated and active components identified and one for plant drug effect is supported by experiments but active components are putatitve or in combination? Botany review work: Taxonomy is important: the plants names have to be written according to the APG IV classification (2016) and add the family name to the species name (s), both in the table and in the text;  current names only in barkets since they are numerous.   the part of the plant (root, leave bark....) where the active component is extract must be shown.

Response: Thank you for your opinion. According to your opinion, the specific active ingredients and compound preparations of botanical medicines are divided into two tables. Meanwhile, according to the APG IV classification (2016), I have added specific family names and plant parts for extracting active ingredients into the T1 table, and changed the official names of some botanical medicines (please see below).

Table1.Targeting and signaling pathways related to megakaryocyte differentiation by botanical drug and their active components in vitro and in vivo

Ingredient

Part used

Botanical name

genus

Research system

Usage and dosage

Mechanism of action

Ref.

PDS-C

Root

Ginseng

Araliaceae

Mice

CHRF-288 cells and Meg01 cells

20, 40 and 80 mg/kg

10, 20, 50 mg/L

Up-regulating the MEK/ERK signaling pathway and transcription factors of GATA1 and RUNX1

[1,2]

PNS

Root

Notoginseng

Araliaceae

K562 cells

Meg01 cells

50 mg/L

Up-regulating the MAPK signaling pathway and transcription factors of GATA1

[3,4]

DYS

Root

S·officinalis.L

Rosaceae

Baf3 cells

32D cells

10 mg/L

By coordinating with stem cell factors (SCFs) and its receptor c-kit to activate its downstream signaling pathway

[5]

DMAG

Mice

HEL cells

5 mg/kg

10, 20 and 40 μM

Activating PI3K/AKT signaling pathway

[6]

TMEA

HEL cells

10 µM and 20 µM

Activating PI3K/AKT signaling pathway and increases ROS levels in cells

[7]

APS

Root

Angelica sinensis (Oliv.) Diels

Apiaceae

Mice

2.5 mg

Activating PI3K/AKT signaling pathway

[8]

Icariin

Aboveground part

Epimedium brevicornum Maxim

Berberidaceae

Mice

10 mg/kg

Inducing the expression of hematopoietic cytokines G-SCF and TPO

[9]

Quercetin

Leaf

Guava

Myrtaceae

White rats

108 mg/kg

Hyaluronidase activity is inhibited, and hyaluronidase can release IL-6 and promote megakaryocyte differentiation

[10]

Orientin

Leaf

Ocimum sanctum

Lamiaceae

Mice

1 mg/kg

EGR1, CXCL12, IL-6ST, Fn1, CCL5 and other genes were up-regulated

[11,12]

Hirsutine

Whole plant

Uncaria rhynchophylla (Miq.) Jacks

Rubiaceae

K562 cells

Meg01 cells

2.5, 5,10,15, and 20 μM

Activating MEK-ERK-FOG1/TAL1 signaling pathway

[13]

Proanthocyanidin A1

Peanut skin

Arachis hypogaea L.

Fabaceae

Mice

Dami cells

25 mg/kg and 50 mg/kg

20μM

Binding to JAK2 activates JAK2/STAT3 pathway and induces megakaryocyte differentiation

[14]

Api7G

Leaf

Canarium subulatum Guill.

Burseraceae

CD34+ cells

5μM

It can enhance the expression of CD41 marker in CD34+ cells

[15,16]

Diosgenin

Root

Dioscorea polystachya Turczaninow

Dioscoreaceae

HEL cells

10μM

Promote the overexpression of p21 and cyclin

[17]

Vasika extract

Leaf

Justicia Adhatado L.

Acanthaceae

Dami cells

10, 20 and 40μg/mL

Enhance the generation of mitochondrial ROS and enhance the permeability of mitochondrial membrane

[18]

Papaya extract

Leaf

Carica papaya

Caricaceae

Wistar rats

200 mg/kg

The expression of CD110 on megakaryocyte surface was increased

[19]

Table2. Plant drug species, associated diseases and prescription sources of the compound in vivo

Compound preparation

Plant name

Diseases

Research system

Usage and dosage

Prescription source

Ref.

DSD

7 kinds of Chinese medicine, such as Angelica sinensis, Cinnamomi ramulus, Paeonia lactiflora Pall., Asarum sieboldii Miq., Tetrapanax papyrifer (Hook.) K. Koch, Ziziphus jujuba Mill. and roasted Glycyrrhiza uralensis Fisch.

Myelosuppression due to chemoradiotherapy

Mice

100、300、900 mg/kg

Traditional Chinese medicine

[20]

FEJ

Equus asinus Linnaeus, Radix Codonopsis Pilosulae, Radix Ginseng Rubra, Fructus Crataegi and Radix Rehmanniae Preparata

Myelosuppression due to chemoradiotherapy

Mice

27.5、55 and 110 mg/kg

Traditional Chinese medicine

[21,22]

SYKT

Sanguis draconis, Radix et rhizoma notoginseng, Radix et rhizoma glycyrrhizae, Radix angelicae sinensis, Zingiber officinale Roscoe, Dioscorea polystachya Turczaninow, Poria cocos(Schw.)Wolf and Amomum villosum Lour.

Myelosuppression due to chemoradiotherapy

Mice

1.2 g/kg

Dai nationality in Southwest China

[23]

Wei Gan Li

Cornu Cervi Pantotrichum, Cordyceps sinensis (BerK.) Sacc., Herba Epimedii, Curcuma phaeocaulis Valeton, Cnidium monnieri (L.) Cuss., Radix Ginseng, Radix Astragali and Radix Glycyrrhizae

Myelosuppression due to chemoradiotherapy

Mice

100 mg/mL、50 mg/mL and 25 mg/mL

Traditional Chinese medicine

[24]

VITA PLAT

Carica papaya, Cissampelospareira, Boerhaviadiffusa, Tinosporacordifolia, Azadirachtaindica

Thrombocytopenia due to dengue fever

Wistar rats

200 and 400 mg/kg

Traditional Indian medicine

[25]

References:

  1. Sun, X.; Zhao, Y.N.; Qian, S.; Gao, R.L.; Yin, L.M.; Wang, L.P.; Chong, B.H.; Zhang, S.Z. Ginseng-Derived Panaxadiol Saponins Promote Hematopoiesis Recovery in Cyclophosphamide-Induced Myelosuppressive Mice: Potential Novel Treatment of Chemotherapy-Induced Cytopenias. Chin J Integr Med 2018, 24, 200-206, doi:10.1007/s11655-017-2754-8.
  2. Wen, W.W.; Sun, X.; Zhuang, H.F.; Lin, X.J.; Zheng, Z.Y.; Gao, R.L.; Yin, L.M. Effects of panaxadiol saponins component as a new Chinese patent medicine on proliferation, differentiation and corresponding gene expression profile of megakaryocytes. Chin J Integr Med 2016, 22, 28-35, doi:10.1007/s11655-015-1970-3.
  3. Gao, R.L.; Chen, X.H.; Lin, X.J.; Qian, X.D.; Xu, W.H.; Chong, B.H. Effects of notoginosides on proliferation and upregulation of GR nuclear transcription factor in hematopoietic cells. Acta Pharmacol Sin 2007, 28, 703-711, doi:10.1111/j.1745-7254.2007.00551.x.
  4. Sun, X.; Gao, R.L.; Lin, X.J.; Xu, W.H.; Chen, X.H. Panax notoginseng saponins induced up-regulation, phosphorylation and binding activity of MEK, ERK, AKT, PI-3K protein kinases and GATA transcription factors in hematopoietic cells. Chin J Integr Med 2013, 19, 112-118, doi:10.1007/s11655-012-1306-4.
  5. Dai, Y.P.; Gao, X.P.; Wu, J.M.; L, X.; Huang, F.H.; Zhou, W.J. Effect of total saponins from Sanguisorba officinalis on megakaryocyte progenitor cells proliferation, diffrentiation and relative receptor expression. China Academic Journal Electronic Publishing House 2014, 39, 1685-1689, doi:10.4268/cjcmm20140928.
  6. Lin, J.; Zeng, J.; Liu, S.; Shen, X.; Jiang, N.; Wu, Y.S.; Li, H.; Wang, L.; Wu, J.M. DMAG, a novel countermeasure for the treatment of thrombocytopenia. Mol Med 2021, 27, 149, doi:10.1186/s10020-021-00404-1.
  7. Li, H.; Jiang, X.; Shen, X.; Sun, Y.; Jiang, N.; Zeng, J.; Lin, J.; Yue, L.; Lai, J.; Li, Y.; et al. TMEA, a Polyphenol in Sanguisorba officinalis, Promotes Thrombocytopoiesis by Upregulating PI3K/Akt Signaling. Front Cell Dev Biol 2021, 9, 708331, doi:10.3389/fcell.2021.708331.
  8. Liu, C.; Li, J.; Meng, F.Y.; Liang, S.X.; Deng, R.; Li, C.K.; Pong, N.H.; Lau, C.P.; Cheng, S.W.; Ye, J.Y.; et al. Polysaccharides from the root of Angelica sinensis promotes hematopoiesis and thrombopoiesis through the PI3K/AKT pathway. BMC Complement Altern Med 2010, 10, 79, doi:10.1186/1472-6882-10-79.
  9. Sun, C.; Yang, J.; Pan, L.; Guo, N.; Li, B.; Yao, J.; Wang, M.; Qi, C.; Zhang, G.; Liu, Z. Improvement of icaritin on hematopoietic function in cyclophosphamide-induced myelosuppression mice. Immunopharmacol Immunotoxicol 2018, 40, 25-34, doi:10.1080/08923973.2017.1392564.
  10. Wiyasihati, S.I.; Wigati, K.W.; Wardani, T. Comparing the effect of red yeast rice,data plam,and guava leaf extract on therombocyte and megakaryocyte count in therombocytopenic white rats. Folia Medica Indonesiana 2013, 49, 82-87.
  11. Bhattacharyya, P.; Bishayee, A. Ocimum sanctum Linn. (Tulsi): an ethnomedicinal plant for the prevention and treatment of cancer. Anticancer Drugs 2013, 24, 659-666, doi:10.1097/CAD.0b013e328361aca1.
  12. Yadav, M.; Song, F.; Huang, J.; Chakravarti, A.; Jacob, N.K. Ocimum flavone Orientin as a countermeasure for thrombocytopenia. Sci Rep 2018, 8, 5075, doi:10.1038/s41598-018-23419-x.
  13. Kang, Y.; Lin, J.; Wang, L.; Shen, X.; Li, J.; Wu, A.; Yue, L.; Wei, L.; Ye, Y.; Yang, J.; et al. Hirsutine, a novel megakaryopoiesis inducer, promotes thrombopoiesis via MEK/ERK/FOG1/TAL1 signaling. Phytomedicine 2022, 102, 154150, doi:10.1016/j.phymed.2022.154150.
  14. Wang, R.; Hu, X.; Wang, J.; Zhou, L.; Hong, Y.; Zhang, Y.; Xiong, F.; Zhang, X.; Ye, W.C.; Wang, H. Proanthocyanidin A1 promotes the production of platelets to ameliorate chemotherapy-induced thrombocytopenia through activating JAK2/STAT3 pathway. Phytomedicine 2022, 95, 153880, doi:10.1016/j.phymed.2021.153880.
  15. Samet, I.; Han, J.; Jlaiel, L.; Sayadi, S.; Isoda, H. Olive (Olea europaea) leaf extract induces apoptosis and monocyte/macrophage differentiation in human chronic myelogenous leukemia K562 cells: insight into the underlying mechanism. Oxid Med Cell Longev 2014, 2014, 927619, doi:10.1155/2014/927619.
  16. Samet, I.; Villareal, M.O.; Motojima, H.; Han, J.; Sayadi, S.; Isoda, H. Olive leaf components apigenin 7-glucoside and luteolin 7-glucoside direct human hematopoietic stem cell differentiation towards erythroid lineage. Differentiation 2015, 89, 146-155, doi:10.1016/j.diff.2015.07.001.
  17. Cailleteau, C.; Micallef, L.; Lepage, C.; Cardot, P.J.; Beneytout, J.L.; Liagre, B.; Battu, S. Investigating the relationship between cell cycle stage and diosgenin-induced megakaryocytic differentiation of HEL cells using sedimentation field-flow fractionation. Anal Bioanal Chem 2010, 398, 1273-1283, doi:10.1007/s00216-010-4062-4.
  18. Gutti, U.; Komati, J.K.; Kotipalli, A.; Saladi, R.G.V.; Gutti, R.K. Justicia adhatoda induces megakaryocyte differentiation through mitochondrial ROS generation. Phytomedicine 2018, 43, 135-139, doi:10.1016/j.phymed.2018.04.038.
  19. Nandini, C.; Madhunapantula, S.V.; Bovilla, V.R.; Ali, M.; Mruthunjaya, K.; Santhepete, M.N.; Jayashree, K. Platelet enhancement by Carica papaya L. leaf fractions in cyclophosphamide induced thrombocytopenic rats is due to elevated expression of CD110 receptor on megakaryocytes. J Ethnopharmacol 2021, 275, 114074, doi:10.1016/j.jep.2021.114074.
  20. Jin, Y.; Qu, C.; Tang, Y.; Pang, H.; Liu, L.; Zhu, Z.; Shang, E.; Huang, S.; Sun, D.; Duan, J.A. Herb pairs containing Angelicae Sinensis Radix (Danggui): A review of bio-active constituents and compatibility effects. J Ethnopharmacol 2016, 181, 158-171, doi:10.1016/j.jep.2016.01.033.
  21. Liu, M.; Tan, H.; Zhang, X.; Liu, Z.; Cheng, Y.; Wang, D.; Wang, F. Hematopoietic effects and mechanisms of Fufang ejiao jiang on radiotherapy and chemotherapy-induced myelosuppressed mice. J Ethnopharmacol 2014, 152, 575-584, doi:10.1016/j.jep.2014.02.012.
  22. Zhang, Y.; Ye, T.; Hong, Z.; Gong, S.; Zhou, X.; Liu, H.; Qian, J.; Qu, H. Pharmacological and transcriptome profiling analyses of Fufang E'jiao Jiang during chemotherapy-induced myelosuppression in mice. J Ethnopharmacol 2019, 238, 111869, doi:10.1016/j.jep.2019.111869.
  23. Chen, T.; Shen, H.M.; Deng, Z.Y.; Yang, Z.Z.; Zhao, R.L.; Wang, L.; Feng, Z.P.; Liu, C.; Li, W.H.; Liu, Z.J. A herbal formula, SYKT, reverses doxorubicininduced myelosuppression and cardiotoxicity by inhibiting ROSmediated apoptosis. Mol Med Rep 2017, 15, 2057-2066, doi:10.3892/mmr.2017.6272.
  24. Chen, Y.; Zhu, B.; Zhang, L.; Yan, S.; Li, J. Experimental study of the bone marrow protective effect of a traditional Chinese compound preparation. Phytother Res 2009, 23, 823-826, doi:10.1002/ptr.2678.
  25. Sailor, G.; Hirani, K.; Parmar, G.; Maheshwari, R.; Singh, R.; Kumar Seth, A. Platelet Augmentation Potential of Polyherbal Formulation in Cyclophosphamide-Induced Thrombocytopenia in Wistar Rats. Folia Med (Plovdiv) 2021, 63, 67-73, doi:10.3897/folmed.63.e49167.
  26. The study focused mainly on chinese plants (plants that could also be found either in the world as sanguisorba sp.) is there non Chinese species that could be of interest?

Response: Thanks for your suggestion. We have consulted the literature and found Carica papaya [1] (Caricaceae family) from Sri Lanka, India and other countries. It has been proved that the leaves and seeds of papaya can improve thrombocytopenia, and the crude extract of papaya leaves has been found to promote the differentiation and maturation of megakaryocytes and release platelets. Api7G [2], an active ingredient in olive leaves from Vietnam and Thailand, was found to induce differentiation of hematopoietic stem cells into megakaryocytes. Diosgenin [3], an active component extracted from the seeds of fenugreek from Iraq to northern Pakistan, was found to induce the differentiation of HEL cells and K562 cells into megakaryocytes. A polyherbal formulation (PHF) [4], is composed of Carica papaya, Cissampelospareira, Boerhaviadiffusa, Tinosporacordifolia and Azadirachtaindica from India, has been found to treat thrombocytopenia caused by chemoradiotherapy or dengue fever, and its long-term use may aid in early recovery from disease with low platelet counts. In addition, there are peanuts, citrus, wild yam, rock algae and so on in the review can be found in the world. This review is readable in understanding the use of plant-based drugs in thrombocytopenia or others.

References:

  1. Jayasinghe, C.D.; Ratnasooriya, W.D.; Premakumara, S.; Udagama, P.V. Platelet augmentation activity of mature leaf juice of Sri Lankan wild type cultivar of Carica papaya L: Insights into potential cellular mechanisms. J Ethnopharmacol 2022, 296, 115511, doi:10.1016/j.jep.2022.115511.
  2. Samet, I.; Villareal, M.O.; Motojima, H.; Han, J.; Sayadi, S.; Isoda, H. Olive leaf components apigenin 7-glucoside and luteolin 7-glucoside direct human hematopoietic stem cell differentiation towards erythroid lineage. Differentiation 2015, 89, 146-155, doi:10.1016/j.diff.2015.07.001.
  3. Cailleteau, C.; Micallef, L.; Lepage, C.; Cardot, P.J.; Beneytout, J.L.; Liagre, B.; Battu, S. Investigating the relationship between cell cycle stage and diosgenin-induced megakaryocytic differentiation of HEL cells using sedimentation field-flow fractionation. Anal Bioanal Chem 2010, 398, 1273-1283, doi:10.1007/s00216-010-4062-4.
  4. Sailor, G.; Hirani, K.; Parmar, G.; Maheshwari, R.; Singh, R.; Kumar Seth, A. Platelet Augmentation Potential of Polyherbal Formulation in Cyclophosphamide-Induced Thrombocytopenia in Wistar Rats. Folia Med (Plovdiv) 2021, 63, 67-73, doi:10.3897/folmed.63.e49167.
  5. The section of putative drugs, it has to be strongly reduced and the conclusion section alsothe main key messages will be in the 2 tables with the correct references.

Response: Thanks for your scientific comment. I have deleted some cumbersome expressions in the part of plant medicine and retained some methodological content, hoping that readers can intuitively understand the research of researchers on the promotion of megakaryocyte differentiation and platelet release by plant medicine. Conclusion and prospect: The differentiation process of megakaryocyte and its physiological significance were partially deleted to improve the readability of the review.

  1. There are many English spelling mistakes and unusual style for a science article as "searchers founds " or the authors says. instead of the study by "xxx et al. " or the team of xxxx: more personnalisation is needed and more scientific style is needed.

Response: Thanks for your scientific comment. I have modified them in lines314 and lines410. We carefully checked the whole manuscript and revised some of words in manuscript. Our English editing have been polished by American Journal Experts. (Abstract line22, 24, 27-28. Introduction line40, 42-46, 50, 53, 56-57, 60-61, 68, 71-76, 81,83-86, 92-93, 101-103. Section 2 line106, 109-110, 116-118, 121-123, 126, 128, 133-141. Section 3 line152, 162, 164, 181-183, 191, 194-195, 197-204. Section 4 line224-230, 240-243, 246-247, 252-257, 261, 270-271, 277,279-290, 292-302, 306-309, 314, 317, 319, 322-326, 332, 336, 340-341, 348-351, 355, 356, 360, 367, 370, 375-376,378, 380-384, 387, 392-393, 399-405, 411-422, 424, 433, 435-436, 439-440, 444-447, 455-456, 464, 468-474, 479, 483-488, 493-495, 505-508, 512, 514, 519-523, 526, 531, 533-537, 543-546, 548-553, 555, 557-560, 565-570, 581, 589-590, 594, 600-601, 604, 606, 613, 617-618. Conclusion and Prospects line631, 633, 641-649,653-665, 669-670, 673-678, 680-692, 698-699, 702-704, 708-711.)

Thank you for all the valuable and helpful comments and suggestions.

Best regards,

Jianming Wu, Ph.D & Professor

Dean of Basic Medical Sciences School &

Head of TCM Pharmacological Lab

Southwest Medical University

Reviewer 2 Report

1. The article is well-written, I enjoyed reading it.

2. Although the authors specified the range of years covered, I believed other information could be obtained in the early 2000 or beyond.

Author Response

Jan. 23, 2023

Dear Expert Reviewer,

Thank you very much for the prompt review process and excellent comments. We greatly appreciate the time and efforts which you have spent on it. We are submitting the revised manuscript entitled “Recent progress in the influence of ethnomedicine on megakaryocyte differentiation” (ID: ijms-2097803) to International Journal of Molecular Sciences.

We have carefully considered your comments and suggestions, and addressed each of the concerns in response to the comments (see point by point response). We have revised the manuscripts based on your comments and carefully checked throughout the manuscript and corrected the language errors. Our point-by-point responses to the comments (in blue) are shown below (in red).

  1. Although the authors specified the range of years covered, I believed other information could be obtained in the early 2000 or beyond.

Response: Thanks for your scientific comment. I have modified it in line 30 and line87.

Thank you for all the valuable and helpful comments and suggestions.

Best regards,

Jianming Wu, Ph.D & Professor

Dean of Basic Medical Sciences School &

Head of TCM Pharmacological Lab

Southwest Medical University

Reviewer 3 Report

Yang et al present a review on botanical drugs and compounds that affect megakaryocyte differentiation and platelet count or improve myelosuppression. The review is significant because traditional preparations have potentially beneficial effects and could improve conditions such as thrombocytopenia and myelosuppression. Improvement in clarity and specifically cited references should be considered.
Specific comments:
I would recommend careful reading and proofreading of the English by a native speaker to improve the clarity of the written text.
The title should be changed as the paper is not about " progress in the influence" - rather it is the review on "the usage of ethnomedicine in modulating megakaryocyte differentiation and platelet counts.
Abstract:
Line 33: "reliable bases" - reliable should be omitted, instead "potentially" should be added later in the sentence "botanical drugs potentially treating..." - it is difficult to draw firm conclusions in the absence of controlled clinical trials confirmed by in vitro data
Introduction
Line 55: omit the word "cut" as MKs do not cut proplatelets into platelets, but release platelets by shear forces
Lines 64-65, and later paragraph: when discussing therapies to treat thrombocytopenia, please provide references. Is it IL -11 or IL -119? Please correct.
Line 74 and paragraph: correct the names of the TPOR agonists: romiplostim, eltrombAopag. Please go into more detail about the differences in the use of hrTPo compared to TPOR agonists and also include the adverse effects of the agonists and add appropriate references.
Line 84: Splenectomy is used as a treatment in rare circumstances (not as a treatment agent). Please describe under what circumstances splenectomy is performed and provide facts such as in what percentage of patients does splenectomy help correct platelet counts. Please provide references.
Line 96: it is not clear what the authors meant by how, what compounds/cytokines could be used for differentiation therapy.
Paragraph, lines 104-109: did the authors mean -bias here at MK? Needs to be reworded for clarity.
Paragraph, lines 140-159: references are missing. Mention what M1, M3 are
Lines 170-171: sentence "We found that ERK1/2..." is unclear, please rewrite
Page 8: Lines 223-233: it would be easier to read and follow if Fig. 1 and Fig. 4 were merged and if an explanation of the basis of the subdivisions was provided in the next sections
Lines 262-234: there is a missing reference
Line 316-321: What is the evidence for the effect of DYS on MKs, please provide appropriate references
Line 356: Please describe what kind of preparation DSD is, and what does it contain?
Line 539-542: the sentence " When there is loss or reduction of Mk cell lines, other cell lines of the same level..." - are the authors thinking about cell lineages in the body? Please reword and use an appropriate reference. Also line 545 "...and K562 cells happened to apoptosis..." if cells undergo apoptosis, how do they differentiate into MKs?
Conclusion and prospects: Most of this section is basically a repetition of the previous parts (introduction of Mks and later compounds), so it could be significantly shortened and focused on perspectives. Line 635: The authors state that "Traditional botanical drugs and compound preparations are widely used in clinical practice..." - please define where exactly (in which states) these drugs and preparations are used to treat myelosuppression, as I do not think these options are considered in most Western countries.
Minor:
Check reference 14, names missing
Line 77: "platelets" instead of "plates

Author Response

Jan. 23, 2023

Dear Expert Reviewer,

Thank you and the reviewers very much for the prompt review process and excellent comments. We greatly appreciate the time and efforts that the International Journal of Molecular Sciences Editorial Team and the reviewers have spent on it. We are submitting the revised manuscript entitled “Recent progress in the influence of ethnomedicine on megakaryocyte differentiation” (ID: ijms-2097803).

We have carefully considered your comments and suggestions, and addressed each of the concerns in response to the comments (see point by point response). We have revised the manuscripts based on your comments and carefully checked throughout the manuscript and corrected the language errors. Our point-by-point responses to the comments (in blue) are shown below (in red).

  1. I would recommend careful reading and proofreading of the English by a native speaker to improve the clarity of the written text.

Response: Thanks for your scientific comment. We carefully checked the whole manuscript and revised some of words in manuscript. Our English editing have been polished by American Journal Experts. (Abstract line22, 24, 27-28. Introduction line40, 42-46, 50, 53, 56-57, 60-61, 68, 71-76, 81,83-86, 92-93, 101-103. Section 2 line106, 109-110, 116-118, 121-123, 126, 128, 133-141. Section 3 line152, 162, 164, 181-183, 191, 194-195, 197-204. Section 4 line224-230, 240-243, 246-247, 252-257, 261, 270-271, 277,279-290, 292-302, 306-309, 314, 317, 319, 322-326, 332, 336, 340-341, 348-351, 355, 356, 360, 367, 370, 375-376,378, 380-384, 387, 392-393, 399-405, 411-422, 424, 433, 435-436, 439-440, 444-447, 455-456, 464, 468-474, 479, 483-488, 493-495, 505-508, 512, 514, 519-523, 526, 531, 533-537, 543-546, 548-553, 555, 557-560, 565-570, 581, 589-590, 594, 600-601, 604, 606, 613, 617-618. Conclusion and Prospects line631, 633, 641-649,653-665, 669-670, 673-678, 680-692, 698-699, 702-704, 708-711.)

  1. The title should be changed as the paper is not about " progress in the influence" - rather it is the review on "the usage of ethnomedicine in modulating megakaryocyte differentiation and platelet counts"

Response: Thanks for your scientific comment. We have thought over your suggestion carefully and decided to adopt it. Change the title to "the application of ethnomedicine in modulating megakaryocyte differentiation and platelet counts".

  1. The details in the Abstract need to be modified

Response: Thanks for your scientific comment.

Line33: I have modified it in line 33 and made a change to the last sentence.

  1. The details in the Introduction need to be modified

Response: Thanks for your scientific comment.

Line55: I have modified it in line 55. Due to the modification of the introduction, I transferred the process of platelet release by megakaryocytes to the section (2), and depicted the process of platelet release in blood vessels in Figure 3B.

We looked at the literature. Cytokine IL-11 stimulates platelet production by increasing the number of hematopoietic stem cells and blood cells. However, there are some adverse reactions such as arrhythmia and acute pulmonary edema [1].

Line74: We have changed the correct name of the TPO receptor agonist. Thrombopoietin receptor agonists currently include first-generation recombinant human thrombopoietin (RhTPO) and second-generation agents (TPO-RAs) such as romiplostim and eltrombAopag. On the one hand, they promote megakaryocyte differentiation and platelet production in vivo, increase the number of peripheral blood platelets. On the other hand, RhTPO may cross-react with endogenous TPO of patients to neutralize autoantibodies, which may result immunogenic reactions. In TPO-RAs, especially romiplostim, which is the only TPO-RA binding with endogenous TPO at the same site, without result in immunogenic reaction. However, there is a risk of thrombosis and up to 10% discontinue reaction. Both require regular treatment and frequent monitoring [2-4].

Line84: Splenectomy is a treatment method used to treat platelets under special circumstances, which has certain physical requirements for patients. In order to enter the focus of the review more quickly, I chose to delete it.

Line96:In fact, this paragraph is intended to introduce a treatment method that can inhibit the abnormal proliferation of tumor cells and treat leukemia by inducing hematopoietic stem cells such as K562 to differentiate into normal megakarnuclei specifically. Considering that the content of this paragraph is really too sudden, it has been introduced in Line 56 that leukemia has been added to the diseases related to thrombocytopenia. Paragraph 4 described the treatment of leukemia with plant drugs and explained differentiation therapy, hoping to better express that differentiation therapy is a new treatment strategy for leukemia.

References:

  1. Nguyen, A.; Repesse, Y.; Ebbo, M.; Allenbach, Y.; Benveniste, O.; Vallat, J.M.; Magy, L.; Deshayes, S.; Maigne, G.; de Boysson, H.; et al. IVIg increases interleukin-11 levels, which in turn contribute to increased platelets, VWF and FVIII in mice and humans. Clin Exp Immunol 2021, 204, 258-266, doi:10.1111/cei.13580.
  2. Ghanima, W.; Cooper, N.; Rodeghiero, F.; Godeau, B.; Bussel, J.B. Thrombopoietin receptor agonists: ten years later. Haematologica 2019, 104, 1112-1123, doi:10.3324/haematol.2018.212845.
  3. Gilreath, J.; Lo, M.; Bubalo, J. Thrombopoietin Receptor Agonists (TPO-RAs): Drug Class Considerations for Pharmacists. Drugs 2021, 81, 1285-1305, doi:10.1007/s40265-021-01553-7.
  4. Al-Samkari, H.; Soff, G.A. Clinical challenges and promising therapies for chemotherapy-induced thrombocytopenia. Expert Rev Hematol 2021, 14, 437-448, doi:10.1080/17474086.2021.1924053.
  5. The details in the Paragraph need to be modified

Response: Lines104-109: In the hematopoietic microenvironment, megakaryocytes and hematopoietic stem cells (VWF+HSC), share the same transcription factors and cytokines. It was proved that megakaryocytes could induce the differentiation of VWF+HSC into megakaryocytes and generate platelets. onsidering that the content of this paragraph is really too sudden. In addition, the environment of megakaryocytes has been described in Lines 128-133. I have merged Lines 104-109 with Lines 183-187 in lines 183-187.

Lines140-159: Thank you for the reminder, MPP1 is classified as a short-term HSC. MPP3 is largely granulocyte/macrophage biased [1].

Lines170-171: Thanks for your reminding, we have modified this sentence

Lines223-233: Thanks to your suggestion, Thanks for your suggestion, we retained Figure 1A and Figure 1C. Figure 1B and Figure 4 were combined and described in the first paragraph of Section 4: Plant drugs can promote megakaryocyte differentiation and release platelets, treat thrombocytopenia and leukemia, and improve myelosuppression and other diseases. Therefore, we classify botanical drugs into three major categories according to the classification of diseases (please see the attachment WORD file).

Lines226-234: Thank you for your reminding. We have provided the reference in the corresponding position.

Lines316-321: Thanks to your suggestion. DYS regulates the proliferation and differentiation of megakaryocytes by coordinating with stem cell factors (SCFs) and its receptor c-kit to activate its downstream signaling pathway [2].

Lines356: Thank you for your reminding, Danggui Sini Decoction (DSD) is a classic recipe for relieving pain, blood deficiency, cold coagulation and other symptoms. DSD consists of 7 kinds of Chinese medicine, such as Angelica sinensis and Paeonia lactiflora. It was found that DSD could significantly increase the proportion of CD34+ cells in mice with bone marrow suppression, and also significantly increase the mRNA expression level of TPO in the spleen of mice [3].

Lines539-542: HL-60 cells, as a kind of dual-energy progenitor cells, can differentiate into granulocytes and macrophages. This paper found that HL-60 cells can differentiate into megakaryocytes through in vitro experiments, which is not only a discovery of a new differentiation pathway, but also provides evidence for the connection of origin cells between HL-60 cells and megakaryocytes. There is no denying that the discovery of this paper has certain potential research value. When I reviewed the literature, I was also curious about this reason for AML transdifferentiation, but unfortunately, no studies have shown that there is a direct correlation between the two types of cells. In addition, this paper only conducted in vitro studies and did not conduct relevant experiments on animals, which should not make people think whether the transdifferentiation pathway can be realized in vivo environment. Thank you for your suggestion. Now we use more objective words to describe this research result. We expect more researchers to read this paper, conduct more in-depth research on the topic of transdifferentiation of different hematopoietic lineages, and look forward to the discovery of more differentiation inducers, which will benefit patients with different types of leukemia [4].

Line545: K562 has the potential to differentiate into erythroid and megakaryoid systems. Under the treatment of botanical drug differentiation, the differentiation of K562 is accompanied by the loss of its proliferative ability, such as G0/G1 cell cycle arrest. The generated megakaryocytes were not platelet-producing and disappeared on day 5 of the experiment. Therefore, atrophy and apoptosis occurred in K562 cells on the 6th day of the experiment [5].

References:

  1. Cabezas-Wallscheid, N.; Klimmeck, D.; Hansson, J.; Lipka, D.B.; Reyes, A.; Wang, Q.; Weichenhan, D.; Lier, A.; von Paleske, L.; Renders, S.; et al. Identification of regulatory networks in HSCs and their immediate progeny via integrated proteome, transcriptome, and DNA methylome analysis. Cell Stem Cell 2014, 15, 507-522, doi:10.1016/j.stem.2014.07.005.
  2. Dai, Y.P.; Gao, X.P.; Wu, J.M.; L, X.; Huang, F.H.; Zhou, W.J. Effect of total saponins from Sanguisorba officinalis on megakaryocyte progenitor cells proliferation, diffrentiation and relative receptor expression. China Academic Journal Electronic Publishing House 2014, 39, 1685-1689, doi:10.4268/cjcmm20140928.
  3. Chen, Q.Q.; Han, X.; Wang, W.M.; Zhao, L.; Chen, A. Danggui sini decoction ameliorates myelosuppression in animal model by upregulating Thrombopoietin expression. Cell Biochem Biophys 2015, 71, 945-950, doi:10.1007/s12013-014-0291-z.
  4. Hou, W.; Wang, Z.Y.; Lin, J.; Chen, W.M. Induction of differentiation of the acute myeloid leukemia cell line (HL-60) by a securinine dimer. Cell Death Discov 2020, 6, 123, doi:10.1038/s41420-020-00354-3.
  5. Hou, W.; Wang, Z.Y.; Lin, J.; Chen, W.M. Induction of differentiation of the acute myeloid leukemia cell line (HL-60) by a securinine dimer. Cell Death Discov 2020, 6, 123, doi:10.1038/s41420-020-00354-3.
  6. Most of this section is basically a repetition of the previous parts (introduction of Mks and later compounds), so it could be significantly shortened and focused on perspectives.

Response: Thanks for your scientific comment. After reviewing the review, we omitted a large number of descriptions of megakaryocyte morphology, differentiation process, and specific regulatory factors, cytokines, and signaling pathways during differentiation.

  1. Line 635: The authors state that "Traditional botanical drugs and compound preparations are widely used in clinical practice..." - please define where exactly (in which states) these drugs and preparations are used to treat myelosuppression, as I do not think these options are considered in most Western countries.

Response: Thanks for your scientific comment. The compound preparations have been summarized in Table2 and the specific countries used have been specified. Natural medicines have been modified and successfully treated for a variety of diseases. According to WHO, natural medicines are widely used in many countries for medical care, disease prevention and treatment, etc. Their research and development time is short, research and development costs are cheap, and compared with synthetic drugs, they are less limited by side effects [1]. In addition to autoimmune defects, thrombocytopenia is also caused by many diseases. Although synthetic drugs have been developed, many indications remain in preclinical research due to adverse reactions and other reasons [2]. Currently, platelet infusion, as the main therapeutic method to improve the adverse reactions caused by thrombocytopenia, is limited by the use of unstable donor, short shelf life and easy contamination [3]. In the face of high drug prices and unstable platelet supply, especially in tropical areas such as Africa to improve the thrombocytopenia caused by malaria, dengue fever and other related diseases, plant drugs are often used to treat thrombocytopenia [4]. With a long history of plant drugs, part of the compound preparation has been widely used in Southeast Asia and Africa. Therefore, I think it is necessary and readable to review the treatment of thrombocytopenia by promoting megakaryocyte differentiation (please see below).

Compound preparation

Plant name

Diseases

Research system

Usage and dosage

Prescription source

Ref.

DSD

7 kinds of Chinese medicine, such as Angelica sinensis, Cinnamomi ramulus, Paeonia lactiflora Pall., Asarum sieboldii Miq., Tetrapanax papyrifer (Hook.) K. Koch, Ziziphus jujuba Mill. and roasted Glycyrrhiza uralensis Fisch.

Myelosuppression due to chemoradiotherapy

Mice

100、300、900 mg/kg

Traditional Chinese medicine

[5]

FEJ

Equus asinus Linnaeus, Radix Codonopsis Pilosulae, Radix Ginseng Rubra, Fructus Crataegi and Radix Rehmanniae Preparata

Myelosuppression due to chemoradiotherapy

Mice

27.5、55 and 110 mg/kg

Traditional Chinese medicine

[6,7]

SYKT

Sanguis draconis, Radix et rhizoma notoginseng, Radix et rhizoma glycyrrhizae, Radix angelicae sinensis, Zingiber officinale Roscoe, Dioscorea polystachya Turczaninow, Poria cocos(Schw.)Wolf and Amomum villosum Lour.

Myelosuppression due to chemoradiotherapy

Mice

1.2 g/kg

Dai nationality in Southwest China

[8]

Wei Gan Li

Cornu Cervi Pantotrichum, Cordyceps sinensis (BerK.) Sacc., Herba Epimedii, Curcuma phaeocaulis Valeton, Cnidium monnieri (L.) Cuss., Radix Ginseng, Radix Astragali and Radix Glycyrrhizae

Myelosuppression due to chemoradiotherapy

Mice

100 mg/mL、50 mg/mL and 25 mg/mL

Traditional Chinese medicine

[9]

VITA PLAT

Carica papaya, Cissampelospareira, Boerhaviadiffusa, Tinosporacordifolia, Azadirachtaindica

Thrombocytopenia due to dengue fever

Wistar rats

200 and 400 mg/kg

Traditional Indian medicine

[10]

References:

  1. Atanasov, A.G.; Waltenberger, B.; Pferschy-Wenzig, E.M.; Linder, T.; Wawrosch, C.; Uhrin, P.; Temml, V.; Wang, L.; Schwaiger, S.; Heiss, E.H.; et al. Discovery and resupply of pharmacologically active plant-derived natural products: A review. Biotechnol Adv 2015, 33, 1582-1614, doi:10.1016/j.biotechadv.2015.08.001.
  2. Nguyen, A.; Repesse, Y.; Ebbo, M.; Allenbach, Y.; Benveniste, O.; Vallat, J.M.; Magy, L.; Deshayes, S.; Maigne, G.; de Boysson, H.; et al. IVIg increases interleukin-11 levels, which in turn contribute to increased platelets, VWF and FVIII in mice and humans. Clin Exp Immunol 2021, 204, 258-266, doi:10.1111/cei.13580.
  3. Gilreath, J.; Lo, M.; Bubalo, J. Thrombopoietin Receptor Agonists (TPO-RAs): Drug Class Considerations for Pharmacists. Drugs 2021, 81, 1285-1305, doi:10.1007/s40265-021-01553-7.
  4. Manasa, K.; Soumya, R.; Vani, R. Phytochemicals as potential therapeutics for thrombocytopenia. J Thromb Thrombolysis 2016, 41, 436-440, doi:10.1007/s11239-015-1257-8.
  5. Jin, Y.; Qu, C.; Tang, Y.; Pang, H.; Liu, L.; Zhu, Z.; Shang, E.; Huang, S.; Sun, D.; Duan, J.A. Herb pairs containing Angelicae Sinensis Radix (Danggui): A review of bio-active constituents and compatibility effects. J Ethnopharmacol 2016, 181, 158-171, doi:10.1016/j.jep.2016.01.033.
  6. Liu, M.; Tan, H.; Zhang, X.; Liu, Z.; Cheng, Y.; Wang, D.; Wang, F. Hematopoietic effects and mechanisms of Fufang ejiao jiang on radiotherapy and chemotherapy-induced myelosuppressed mice. J Ethnopharmacol 2014, 152, 575-584, doi:10.1016/j.jep.2014.02.012.
  7. Zhang, Y.; Ye, T.; Hong, Z.; Gong, S.; Zhou, X.; Liu, H.; Qian, J.; Qu, H. Pharmacological and transcriptome profiling analyses of Fufang E'jiao Jiang during chemotherapy-induced myelosuppression in mice. J Ethnopharmacol 2019, 238, 111869, doi:10.1016/j.jep.2019.111869.
  8. Chen, T.; Shen, H.M.; Deng, Z.Y.; Yang, Z.Z.; Zhao, R.L.; Wang, L.; Feng, Z.P.; Liu, C.; Li, W.H.; Liu, Z.J. A herbal formula, SYKT, reverses doxorubicininduced myelosuppression and cardiotoxicity by inhibiting ROSmediated apoptosis. Mol Med Rep 2017, 15, 2057-2066, doi:10.3892/mmr.2017.6272.
  9. Chen, Y.; Zhu, B.; Zhang, L.; Yan, S.; Li, J. Experimental study of the bone marrow protective effect of a traditional Chinese compound preparation. Phytother Res 2009, 23, 823-826, doi:10.1002/ptr.2678.
  10. Sailor, G.; Hirani, K.; Parmar, G.; Maheshwari, R.; Singh, R.; Kumar Seth, A. Platelet Augmentation Potential of Polyherbal Formulation in Cyclophosphamide-Induced Thrombocytopenia in Wistar Rats. Folia Med (Plovdiv) 2021, 63, 67-73, doi:10.3897/folmed.63.e49167.

Thank you for all the valuable and helpful comments and suggestions.

Best regards,

Jianming Wu, Ph.D & Professor

Dean of Basic Medical Sciences School &

Head of TCM Pharmacological Lab

Southwest Medical University

Round 2

Reviewer 1 Report

the reviewed manuscript is greatly improved

some minor spelling and plants name not translated in latin according to the APG classification in the  section 4 needed only very slight revision and it will acceptable

Author Response

Jan. 28, 2023

Dear Expert Reviewer,

Thank you very much for the prompt review process and excellent comments. We greatly appreciate the time and efforts which you have spent on it. We are submitting the revised manuscript entitled “Recent progress in the influence of ethnomedicine on megakaryocyte differentiation” (ID: ijms-2097803) to International Journal of Molecular Sciences.

We have carefully considered your comments and suggestions, and addressed each of the concerns in response to the comments (see point by point response). We have revised the manuscripts based on your comments and carefully checked throughout the manuscript and corrected the language errors. Our point-by-point responses to the comments (in blue) are shown below (in red).

  1. Some minor spelling and plants name not translated in latin according to the APG classification in the section 4 needed only very slight revision and it will acceptable

Reply: Thanks for your scientific comment, which is very helpful to me. In this paper, we have added or modified some Latin names of botanical drugs, such as line239, line277, line281, line292, line294, line357, line410, line441, and revised Latin names of botanical drugs in table1 and table2. (please see below).

Table1.Targeting and signaling pathways related to megakaryocyte differentiation by botanical drug and their active components in vitro and in vivo

Ingredient

Part used

Botanical name

genus

Research system

Usage and dosage

Mechanism of action

Ref.

PDS-C

Root

Panax ginseng C. A. Mey.

Araliaceae

Mice

CHRF-288 cells and Meg01 cells

20, 40 and 80 mg/kg

10, 20, 50 mg/L

Up-regulating the MEK/ERK signaling pathway and transcription factors of GATA1 and RUNX1

[39,40]

PNS

Root

Panax notoginseng (Burkill) F. H. Chen

Araliaceae

K562 cells

Meg01 cells

50 mg/L

Up-regulating the MAPK signaling pathway and transcription factors of GATA1

[44,45]

DYS

Root

Sanguisorba officinalis L.

Rosaceae

Baf3 cells

32D cells

10 mg/L

By coordinating with stem cell factors (SCFs) and its receptor c-kit to activate its downstream signaling pathway

[78]

DMAG

Mice

HEL cells

5 mg/kg

10, 20 and 40 μM

Activating PI3K/AKT signaling pathway

[49]

TMEA

HEL cells

10 µM and 20 µM

Activating PI3K/AKT signaling pathway and increases ROS levels in cells

[51]

APS

Root

Angelica sinensis (Oliv.) Diels

Apiaceae

Mice

2.5 mg

Activating PI3K/AKT signaling pathway

[53]

Icariin

Aboveground part

Epimedium brevicornu Maxim.

Berberidaceae

Mice

10 mg/kg

Inducing the expression of hematopoietic cytokines G-SCF and TPO

[56]

Quercetin

Leaf

Psidium guajava L.

Myrtaceae

White rats

108 mg/kg

Hyaluronidase activity is inhibited, and hyaluronidase can release IL-6 and promote megakaryocyte differentiation

[57]

Orientin

Leaf

Ocimum sanctum L.

Lamiaceae

Mice

1 mg/kg

EGR1, CXCL12, IL-6ST, Fn1, CCL5 and other genes were up-regulated

[63,64]

Hirsutine

Whole plant

Uncaria rhynchophylla (Miq.) Miq. Ex Havil.

Rubiaceae

K562 cells

Meg01 cells

2.5, 5,10,15, and 20 μM

Activating MEK-ERK-FOG1/TAL1 signaling pathway

[65]

Proanthocyanidin A1

Seed cost

Arachis hypogaea Linn.

Fabaceae

Mice

Dami cells

25 mg/kg and 50 mg/kg

20μM

Binding to JAK2 activates JAK2/STAT3 pathway and induces megakaryocyte differentiation

[70]

Api7G

Leaf

Canarium

Album

(Lour.) DC.

Burseraceae

CD34+ cells

5μM

It can enhance the expression of CD41 marker in CD34+ cells

[71,72]

Diosgenin

Root

Dioscorea hispida

Dennst

Dioscoreaceae

HEL cells

10μM

Promote the overexpression of p21 and cyclin

[79]

Vasika extract

Leaf

Justicia Adhatado L.

Acanthaceae

Dami cells

10, 20 and 40μg/mL

Enhance the generation of mitochondrial ROS and enhance the permeability of mitochondrial membrane

[58]

Papaya extract

Leaf

Carica papaya

Caricaceae

Wistar rats

200 mg/kg

The expression of CD110 on megakaryocyte surface was increased

[62]

Table2. Plant drug species, associated diseases and prescription sources of the compound in vivo

Compound preparation

Plant name

Diseases

Research system

Usage and dosage

Prescription source

Ref.

DSD

7 kinds of Chinese medicine, such as Angelica sinensis (Oliv.) Diels, Cinnamomum cassia Nees ex Blume, Paeonia lactiflora Pall., Asarum heterotropoieds F. Schmidt, Tetrapanax papyrifer (Hook.) K. Koch, Zizypgus vulgaris var. intermis Bunge and Glycyrrhiza uralensis Fisch. ex DC.

Myelosuppression due to chemoradiotherapy

Mice

100、300、900 mg/kg

Traditional Chinese medicine

[55]

FEJ

Equus asinus L., Panax ginseng C. A. Mey., Rehmannia glutinosa (Gaertn.) DC., Codonopsis pilosula (Franch.) Nannf. and Crataegus pinnatifida Bunge

Myelosuppression due to chemoradiotherapy

Mice

27.5、55 and 110 mg/kg

Traditional Chinese medicine

[80,81]

SYKT

Zingiber officinale Rosc., Daemonorops draco (Willd.) Blume., Disoscorea opposita Thunb., Poria cocos (Schw.) Wolf, Amomum villosum Lour., Angelica sinensis (Oliv.) Diels, Panax noto-ginseng (Bur-kill) F. H. Chen and Glycyrrhiza uralensis Fisch. ex DC.

Myelosuppression due to chemoradiotherapy

Mice

1.2 g/kg

Dai nationality in Southwest China

 [82]

Wei Gan Li

Cervus nippon Temminck, Astragalus membranaceus Moench, Panax ginseng C. A. Mey., Epimedium brevicornu Maxim., Curculigo orchioides Gaertn., Cnidium monnieri (Linn.) Cuss., Paeonia lutea Delavay ex Franch and Glycyrrhiza uralensis Fisch. ex DC.

Myelosuppression due to chemoradiotherapy

Mice

100 mg/mL、50 mg/mL and 25 mg/mL

Traditional Chinese medicine

[83]

VITA PLAT

Carica papaya, Cissampelos pareira var. hirsute (Buch. Ex DC) Forman, Boerhavia diffusa L., Tinospora cordifolia (Willd.) Miers, and Azadirachta indica A. Juss.

Thrombocytopenia due to dengue fever

Wistar rats

200 and 400 mg/kg

Traditional Indian medicine

[84]

Thank you for all the valuable and helpful comments and suggestions.

Best regards,

Jianming Wu, Ph.D & Professor

Dean of Basic Medical Sciences School &

Head of TCM Pharmacological Lab

Southwest Medical University

Reviewer 3 Report

The authors have answered and accepted most of the suggestions, the review has been improved. 

Author Response

Jan. 28, 2023

Dear Expert Reviewer,

Many thanks for the prompt review process and excellent comments. We greatly appreciate the time and efforts which you have spent on it. We are submitting the revised manuscript entitled “Recent progress in the influence of ethnomedicine on megakaryocyte differentiation” (ID: ijms-2097803) to International Journal of Molecular Sciences.

We have carefully considered your comments and suggestions and revised the manuscripts throughout the manuscript and corrected the language errors such as line239, line277, line281, line292, line294, line357, line410, line441, and revised Latin names of botanical drugs in table1 and table2. We believe that the revised manuscript has been improved and we hope it is now acceptable for publication.

Thank you for all the valuable and helpful comments and suggestions.

Best regards,

Jianming Wu, Ph.D & Professor

Dean of Basic Medical Sciences School &

Head of TCM Pharmacological Lab

Southwest Medical University